# Crowdsourced mapping of unexplored target space of kinase inhibitors

Anna Cichońska [1,2,3,51], Balaguru Ravikumar [1,51], Robert J. Allaway [4,51], Fangping Wan [5], Sungjoon Park[6], Olexandr Isayev [7], Shuya Li[5], Michael Mason [4], Andrew Lamb[4], Ziaurrehman Tanoli [1], Minji Jeon[6], Sunkyu Kim[6], Mariya Popova[7], Stephen Capuzzi [8], Jianyang Zeng [5], Kristen Dang[4], Gregory Koytiger[9], Jaewoo Kang [6], Carrow I. Wells [10], Timothy M. Willson [10], The IDG-DREAM Drug-Kinase Binding Prediction Challenge Consortium*, Tudor I. Oprea [11], Avner Schlessinger [12], David H. Drewry [10], Gustavo Stolovitzky [13], Krister Wennerberg [14,52✉], Justin Guinney[4,52✉] & Tero Aittokallio [1,2,15,16,17,52✉]

Despite decades of intensive search for compounds that modulate the activity of particular protein targets, a large proportion of the human kinome remains as yet undrugged. Effective approaches are therefore required to map the massive space of unexplored compound–kinase interactions for novel and potent activities. Here, we carry out a crowdsourced benchmarking of predictive algorithms for kinase inhibitor potencies across multiple kinase families tested on unpublished bioactivity data. We find the top-performing predictions are based on various models, including kernel learning, gradient boosting and deep learning, and their ensemble leads to a predictive accuracy exceeding that of single-dose kinase activity assays. We design experiments based on the model predictions and identify unexpected activities even for under-studied kinases, thereby accelerating experimental mapping efforts. The open-source prediction algorithms together with the bioactivities between 95 compounds and 295 kinases provide a resource for benchmarking prediction algorithms and for extending the druggable kinome.

A full list of author affiliations appears at the end of the paper.

Only 11% of the human proteome can be currently targeted by small molecules or drugs, whereas one in three proteins remains understudied[1]. Despite many years of target-based drug discovery, chemical agents inhibiting single protein targets are still rare[2]. Most approved drugs have multiple targets, suggesting their therapeutic efficacy as well as adverse side-effects originate from polypharmacological effects[3]. Systematic mapping of the target binding profiles is therefore critical not only to explore the therapeutic potential of promiscuous agents, but also to better predict and manage possible adverse effects within early stages of drug development process to mitigate future risks and costs. Comprehensive understanding of the polypharmacological effects of approved drugs could also uncover novel off-target potencies to extend their therapeutic application area via off-label use or repurposing[4]. However, due to the massive size of the chemical universe, an exhaustive experimental mapping of compound-target activities is infeasible, even with automated high-throughput profiling assays.

To accelerate the mapping efforts, we hosted the IDG-DREAM Drug-Kinase Binding Prediction Challenge, a crowd-sourced competition that evaluated the power of machine learning (ML) models as a systematic and cost-effective means for predicting yet unexplored compound-target potencies. The Challenge focused on predicting quantitative target activities of kinase inhibitors, since kinases are implicated in a wide range of diseases, such as cardiovascular disorders and cancers. However, protein kinase domains are inherently similar in their structure and sequence, and most kinase inhibitors bind to conserved ATP-binding pockets, leading to extensive target promiscuity and polypharmacological effects[5–8]. Such multi-target activities require methods for effective target deconvolution, including multi-target ML approaches, that leverage the information extracted from similar kinases and compounds to predict the activity of so far unexplored compound-kinase interactions[9,10].

The specific questions this Challenge sought to address were: (i) What are the best computational modeling approaches for predicting quantitative compound-target activity profiles?; (ii) What are the best molecular, chemical, and protein descriptors for maximal prediction accuracy?; and (iii) What are the most informative bioactivity assays for dose-response bioactivity prediction? Models submitted to the Challenge were quantitatively evaluated using bioactivity data contributed by—and in partnership with—the Illuminating the Druggable Genome (IDG) consortium (https://druggablegenome.net/). IDG is a NIH Common Fund program aimed at improving our understanding of understudied proteins within three drug-targeted protein families: G-protein coupled receptors, ion channels, and protein kinases[1]. Specifically, it seeks to improve the druggability of dark kinases by kinome-wide profiling small-molecule agents, with the goal of extending the activity information for the understudied human kinome.

Here, we describe the benchmarking results of the Challenge, as well as the post-Challenge analysis of top-performing models to identify so far unexplored kinase inhibitor activities. The benchmarking results include a total of 268 predictions from 212 active Challenge participants, covering a wide range of ML approaches, including linear regularized regression, deep and kernel learning algorithms, and gradient boosting decision trees.

## Results

### Challenge implementation and training datasets
To develop regression models for prediction of quantitative bioactivities, participants were encouraged to utilize a wide variety of bioactivity data for model training and cross-validation through open databases such as ChEMBL[11], BindingDB[12], and IDG Pharos[13]

(Fig. 1). For training data collection, integration, management and harmonization, the Challenge made use of an open-data platform, DrugTargetCommons (DTC)[14]. DTC is a community platform that provides a comprehensive and standardized interface to retrieve compound-target profiles and related information to support predictive activity modeling (Supplementary Fig. 1). The Challenge infrastructure was built on the Synapse collaborative science platform[15], which supported receiving, validating and scoring of the teams' predictions as well as long-term management of the test bioactivity data and submitted Challenge models as a benchmarking resource (Fig. 1).

### Challenge test datasets of kinase inhibitors
The blinded evaluation of the model predictions was based on unpublished kinase activity data generated by the IDG Consortium, with a focus to investigate especially understudied yet readily screenable human kinome, so-called dark kinases[13], and those lacking small-molecule activities in ChEMBL[11], but with a robust assay readily available through commercial vendors[16]. The Challenge was conducted over a series of rounds based on availability of test datasets (Supplementary Fig. 3). Round 1 test dataset was generated based on the two-step screening approach[6,7,16], where the quantitative dose-response measurement of the dissociation constant ($K_d$) activities was carried out across 430 interactions between 70 inhibitors and 199 kinases that had inhibition >80% in the single-dose kinome activity scan (see Methods). An additional set of completely new $K_d$ data was generated for Round 2, consisting of 394 multi-dose assays between 25 inhibitors and 207 kinases with single-dose inhibition >80%. Together, these 824 $K_d$ assays spanned a total of 95 compounds and 295 kinases, covering 57% of the human kinome (Fig. 2a, b). The Challenge test data consisted both of promiscuous compounds targeting multiple kinases at low concentrations, compounds with narrow target profiles, as well as compounds with no potent targets among the tested kinases (Supplementary Fig. 2).

Round 1 enabled the teams to carry out the initial testing of various model classes and data resources, whereas Round 2, implemented 6 months later once the new $K_d$ data became available, was used to score the final prediction models and to select the top-performing teams. None of the $K_d$ values were available in the public domain, and the Round 1 test data remained blinded in Round 2. Round 1 and 2 test datasets had very similar $pK_d$ distributions (Fig. 2c), which provided comparable binding affinity outcome data to monitor the improvements made by the teams between the two rounds. The tested kinase inhibitors in the two test sets were mutually exclusive between the rounds (Fig. 2a), with Round 2 including less selective inhibitors with broader target profiles (Fig. 2d), and therefore fewer inactive compound-kinase pairs ($pK_d = 5$). Round 1 and 2 kinase targets were partly overlapping, and covered all the major kinase families and groups (Fig. 2b, e). Taken together, these two test datasets provided a standardized and sufficiently large quantitative bioactivity resource to evaluate the accuracy of predicting on- and off-target kinase activities, using pharmacologically realistic and computationally rather challenging compound and target spaces of multi-targeting kinase inhibitors.

### Predictive performance of the Challenge models
The competition phase challenged the participants to predict blinded $K_d$ profiles between 95 inhibitors and 295 kinases. Since the goal of this Challenge was to encourage regression model development that would exceed state-of-the-art, we selected as baseline model a recently published and experimentally validated kernel

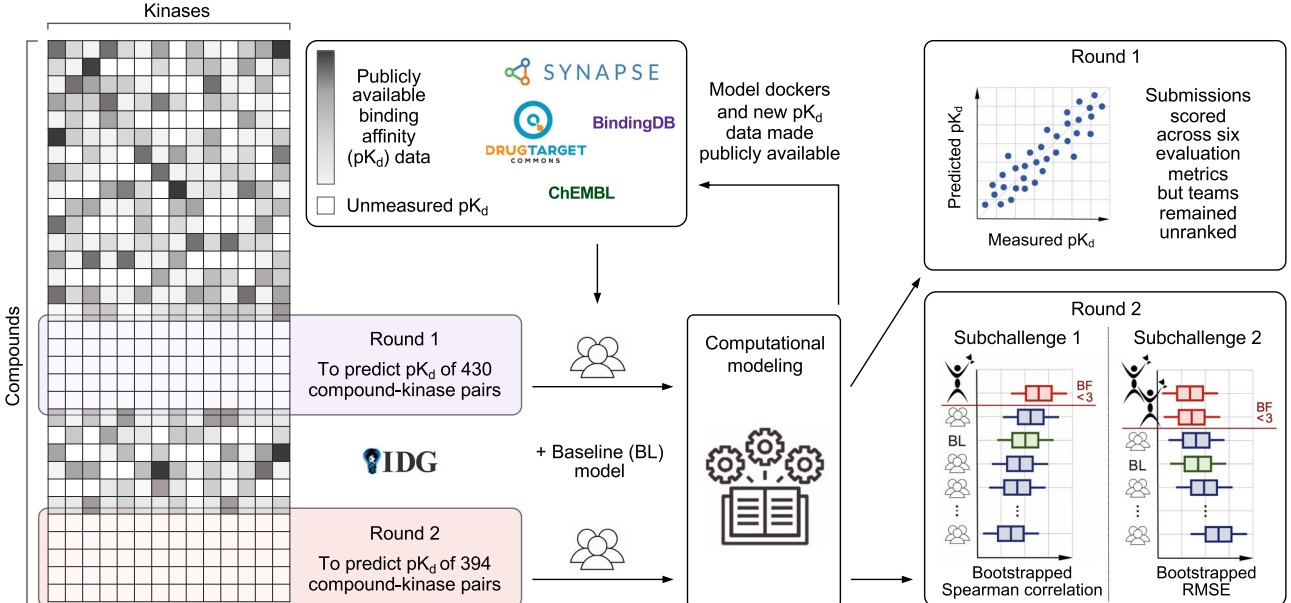

**Fig. 1 Implementation of the IDG-DREAM Drug-Kinase Binding prediction Challenge.** The participants had access to publicly available large-scale target profiling training data, and the quantitative predictions from regression models were then validated in two unpublished and blinded test datasets profiled by the Illuminating the Druggable Genome (IDG) program (Round 1 and Round 2 datasets). Heatmap on the left is for illustrative purposes only (see Supplementary Fig. 2 for the actual test data matrices, and Supplementary Fig. 3 for the Challenge timeline). All the models, new bioactivity data, and benchmarking infrastructure are openly available to support future target prediction and benchmarking studies. BF Bayes factor; RMSE Root Mean Square Error.

regression approach for compound-kinase activity prediction[17]. The performance of the Challenge model predictions improved from Round 1 to Round 2 submissions as measured by Spearman correlation (two-sample Wilcoxon test, $P < 0.005$; Fig. 3a) and Root Mean Square Error (RMSE, $P < 10^{-6}$; Fig. 3c). Comparison against the baseline model indicated that the Round 2 dataset was marginally easier to predict (Supplementary Fig. 4), partly due to a smaller proportion of inactive pairs in Round 2 ($pK_d = 5$, Fig. 2c). To take into account this shift, we compared the submissions against a set of random predictions. Using Spearman correlation, we observed that 48% of the submissions were better than random in Round 1, compared to 61% in Round 2 (Fig. 3b). Using RMSE, 71% of the submissions in Round 1 were better than random, compared to 76% in Round 2 (Fig. 3d).

The 20 teams that participated in both rounds improved their $K_d$ predictions ($P < 0.05$ and $P < 0.001$ for Spearman correlation and RMSE, respectively, paired Wilcoxon signed-rank test), but when comparing against the baseline model, the overall improvements became insignificant ($P > 0.05$). However, there were individual teams (like Zahraa Sobhy) that were able to improve their predictions considerably between the two rounds. The practical upper bound of the model predictions was defined based on experimental replicates of $K_d$ measurements (Fig. 3b, d). The predictive accuracy of the top-performing models in Round 2 was relatively high based on both of the winning metrics, Spearman correlation for ranked pairs predictions and RMSE for quantitative activity predictions; these metrics showed less-correlated performance over the less-accurate models in Round 2 (Fig. 3f). The tie-breaking metric, averaged area under the receiver operating characteristic (ROC) curve, provided complementary information on prediction accuracy when compared to RMSE but not to Spearman correlation (Supplementary Fig. 5). Overall, the models based on deep learning algorithms did not perform better than other learning algorithms submitted in Round 2 (Fig. 3f).

**Selection of the top-performing Challenge models**. The top-performing models were selected in Round 2 based on 394 $pK_d$ predictions between 25 compounds and 207 kinases. Only those participants who submitted their Dockerized models, method write-ups, and method surveys were qualified to win the two sub-challenges (see Supplementary Table 1 for all submissions in Round 2 from the participants who submitted method surveys, together with their model features and training data). To select the top performers, we conducted a bootstrap analysis of each participant's best submission, and then calculated a Bayes factor ($K$) relative to the bootstrapped overall best submission for each winning metric (Supplementary Fig. 6). Considering Spearman correlation, the top performer was team Q.E.D ($K < 3$; Fig. 4a). For the RMSE metric, the top-performing teams were AI Winter is Coming (AIWIC) and DMIS_DK ($K < 3$), with AIWIC having a marginally better tie-breaking metric (average AUC of 0.773; Fig. 4b). Only two non-qualifying participants (Gregory Koytiger and Olivier Labayle) showed comparable performance. Overall, these five teams performed the best across the 54 teams and the 99 total submissions in Round 2 (Supplementary Fig. 7).

Notably, the top-performing models were based on rather different ML approaches, including deep learning, graph convolutional networks, gradient boosting decision trees, kernel learning and regularized regression (Table 1). To study whether combining predictions from the multiple ML approaches could further improve prediction accuracy, we constructed an ensemble model by simple mean aggregation of an increasing number of top-performing models in Round 2. A combination of the four best performing models resulted in the peak Spearman correlation (Fig. 4c), demonstrating a complementary value of these models and their features. After adding more models, the ensemble prediction accuracy decreased rapidly in terms of Spearman correlation and RMSE (Fig. 4d). Combinations of four random models resulted in a decreased performance compared to the top-model ensemble (empirical $P = 0.0$, Supplementary Fig. 8). This suggests that combination of best performing

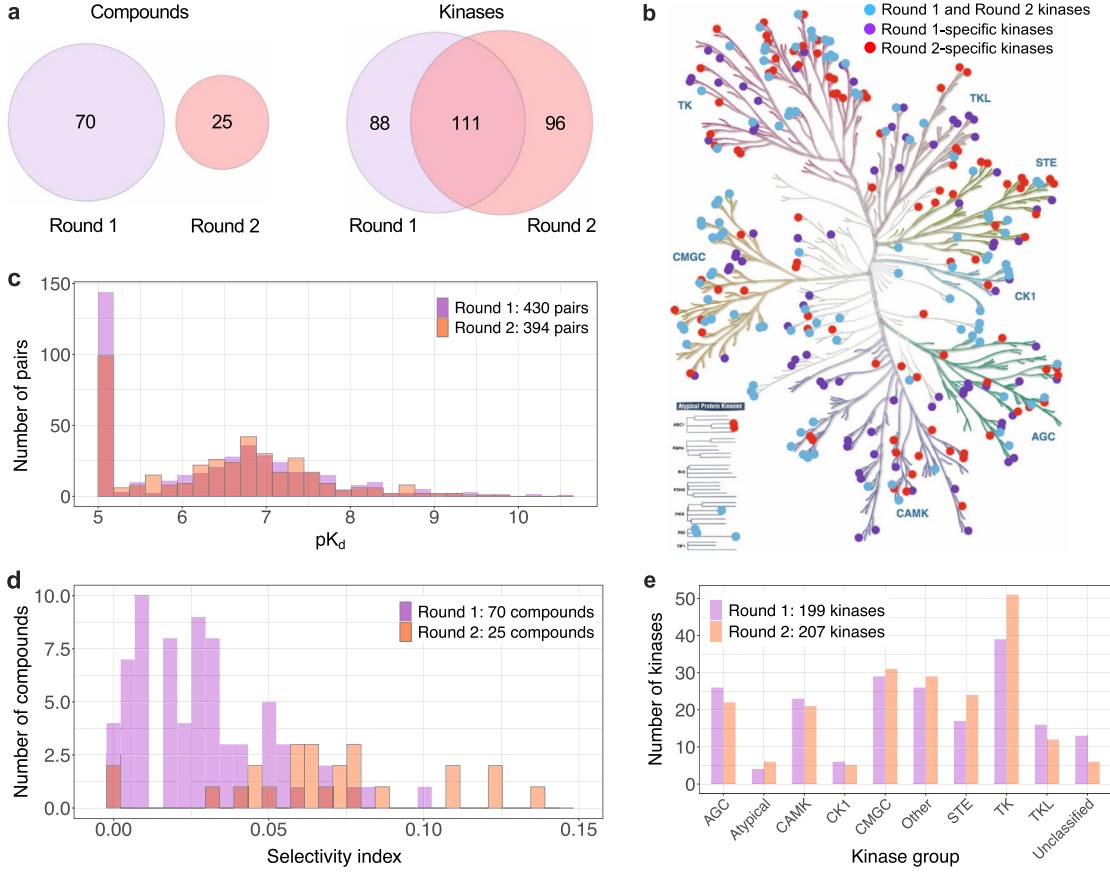

**Fig. 2 Challenge test datasets. a** The overlap between Round 1 and Round 2 kinase inhibitors and kinase targets, and their distributions in the kinome tree (**b**), and across various kinase groups (**e**). **c** The quantitative dissociation constant ($K_d$) of compound-kinase activities was measured in dose-response assays (see Methods), presented in the logarithmic scale as $pK_d = -\log_{10}(K_d)$. The higher the $pK_d$ value, the higher the inhibitory ability of a compound against a protein kinase (Supplementary Data 1 includes the compounds and kinases in Round 1 and Round 2 test datasets). The frequent values of $pK_d = 5$ originate from inactive pairs (maximum tested concentration of 10 µM in the multi-dose activity profiling). **d** The selectivity index of kinase inhibitors was calculated based on the single-dose activity assay (at 1 µM concentration) across the full compound-kinase matrices before the Challenge. The kinome tree figure was created with KinMap, reproduced courtesy of Cell Signaling Technology, Inc. Source data are provided as a Source Data file[54].

approaches using an ensemble model leads to accurate and robust predictions of kinase inhibitor potencies across multiple kinase families.

**Analysis of the Q.E.D and ensemble models**. To better understand how the amount and diversity of training data contribute to the Q.E.D model accuracy, we removed training bioactivity data based on compound structural similarity (Fig. 5a). Surprisingly, we found that the structural similarity of the training and test compounds was relatively unimportant in predicting the activity of the test compounds, indicating that the Q.E.D model made use of other, structurally diverse set of compounds in the test compound activity predictions (Fig. 5a). At the lower similarity cutoffs (Tanimoto similarity <0.7), the model performance decreased substantially, likely due to an increased disparity in chemistry between the test and training compounds, as well as an overall decrease in the training dataset size. We also performed a similar experiment to test the importance of high- and low-potency compounds on model accuracy (Fig. 5b), by removal of training data compounds with high $pK_d$, low $pK_d$, or both. As anticipated, we observed that removal of high $pK_d$ compound-kinase pairs ($pK_d$ values larger than 8) reduced performance of the model. This is likely a consequence of both loss of the overall number of training data and loss of rare extreme activities. However, removal of the small number of compound-kinase pairs with the

most extreme $pK_d$ values (training on $pK_d$ values between 4 and 10) had no effect on accuracy.

We further systematically investigated the relative contributions of various chemical and protein descriptors to the predictive performance of the Q.E.D model. These results showed that whilst several different chemical fingerprints performed similarly well (Supplementary Fig. 10), the choice of protein descriptor had a more notable impact on the model prediction accuracy (Fig. 6a). Especially the protein kernel based on amino acid subsequences of ATP-binding pockets resulted in a poor performance (adjusted $P < 10^{-10}$, Pearson and Filon test), compared to the full amino acid sequences, which can at least partly be explained by the missing subsequences for several kinases that reduced the training dataset size and also led to some activity predictions of zero (Supplementary Fig. 11; we note that this is also the case for kinase domain sequences). We also re-trained the Q.E.D model with different combinations of training bioactivity data types to investigate which types contributed most to the high prediction accuracy. We observed that while $K_d$ alone or in combination with other bioactivity data types, especially with $K_i$, systematically resulted in rather accurate $K_d$ predictions, the other types led to significantly worse prediction performances (Fig. 6b). Especially the rather abundant $EC_{50}$ and $IC_{50}$ bioactivities alone led to poor $pK_d$ prediction accuracy (Supplementary Fig. 12). This result can be explained by the fact that, in contrast to $K_d$ affinity assay, $EC_{50}$

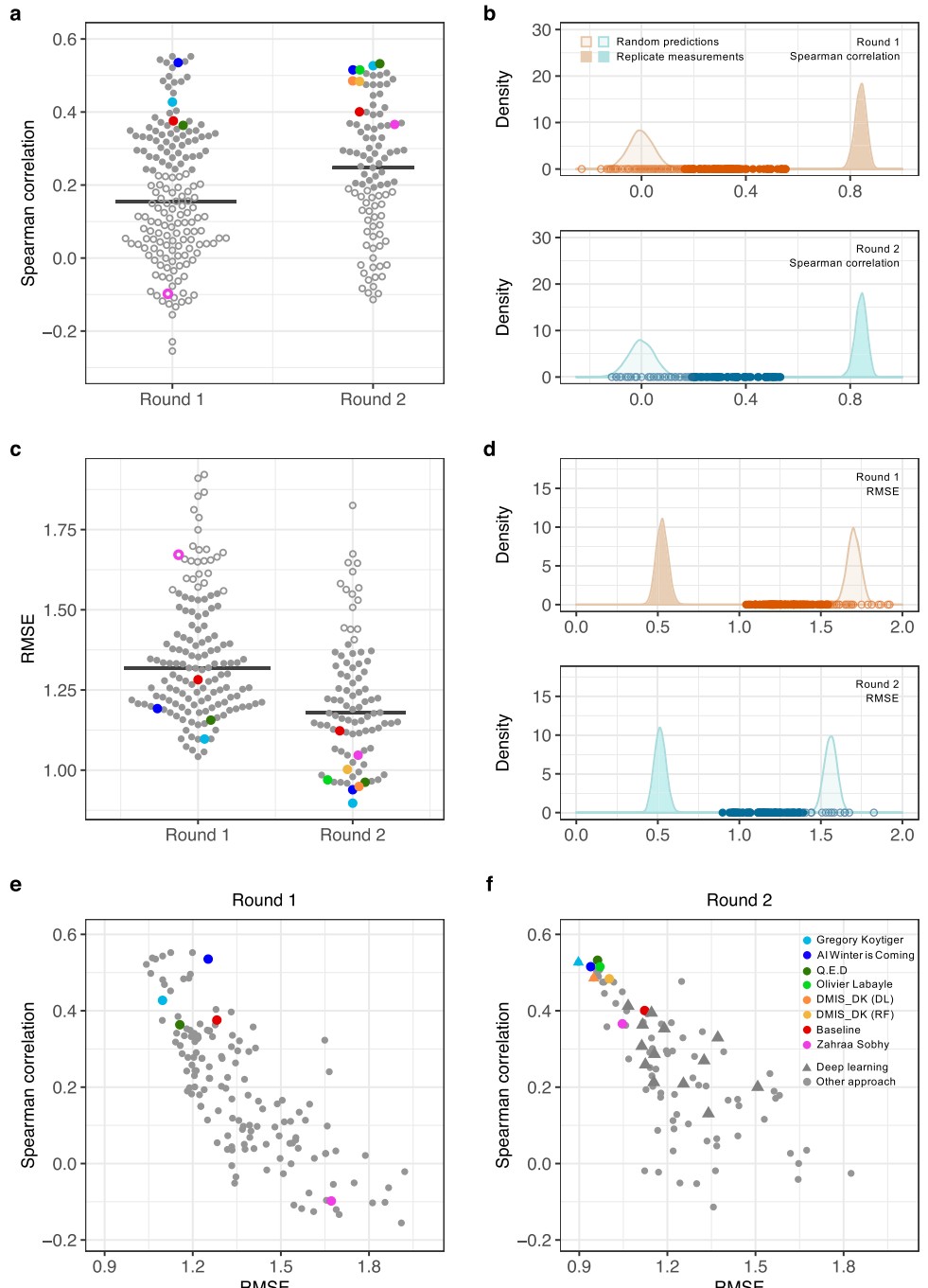

**Fig. 3 Overall performance of the Challenge submissions. a**, **c** Performance of the submissions in terms of the two winning metrics in Round 1 (*n* = 169 submissions) and Round 2 (*n* = 99 submissions). The horizontal lines indicate median correlation and the colors mark the baseline model and the top-performing participants in Round 2 (see the color legend of **f**). The empty circles mark the submissions that did not differ from random predictions (the open pink circle indicates the Round 1 submission of Zahraa Sobhy as an example). The baseline model[17] remained the same in both of the rounds. **b**, **d** Distributions of the random predictions (based on 10,000 permuted pK$_d$ values) and replicate distributions (based on 10,000 subsamples with replacement of overlapping pK$_d$ pairs between two large-scale target activity profiling studies[5,6]) in Round 1 (top panel) and Round 2 (bottom). The points correspond to the individual submissions. **e**, **f** Relationship of the two winning metrics across the submissions in Round 1 and Round 2. The triangle shape indicates submissions based on deep learning (DL) in Round 2 (**f**). For instance, team DMIS_DK submitted predictions based both on random forest (RF) and DL algorithms in Round 2, where the latter showed slightly better accuracy. A total of 33 submissions with Root Mean Square Error (RMSE) >2 are omitted in the RMSE results (**c**, **e**, **f**). Source data are provided as a Source Data file[54].

and IC$_{50}$ values are dependent on the pre-specified target protein concentration of the assay.

We also investigated how well the Challenge models predicted various kinase classes to study their applicability ranges. We first ranked the compound-kinase pairs based on their absolute errors

(AEs), and then systematically explored whether any kinase group or family would be enriched among the best or worst-predicted pairs (see Methods). When considering 90 out of 99 Challenge submissions in Round 2 (with average AE < 2), the compound-kinase pairs involving mitogen-activated protein (MAP) and

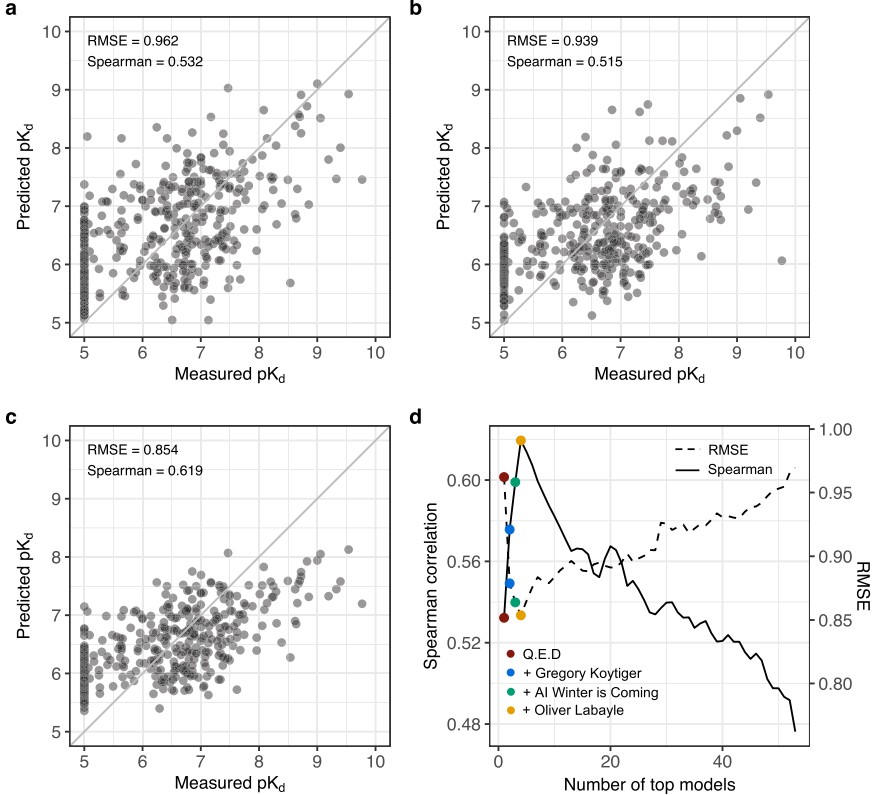

**Fig. 4 The top-performing Challenge models and their ensemble combination. a** Spearman correlation sub-challenge top performer in Round 2 (Q.E.D). **b** RMSE sub-challenge top performer in Round 2 (AI Winter is Coming). The points correspond to 394 pairs between 25 compounds and 207 kinases. **c** Ensemble model that combines the top four models selected based on their Spearman correlation in Round 2. **d** The mean aggregation ensemble model was constructed by adding an increasing number of top-performing models based on their Spearman correlation (the solid curve), until the ensemble correlation dropped below 0.45. The peak performance was reached after aggregating four teams (marked in the legend; see Supplementary Fig. 9 for all the teams. Note: ensemble prediction from a total of 21 best teams had a significantly better Spearman correlation compared to the Q.E.D model alone). The right-hand y-axis and the dotted curve show the Root Mean Square Error (RMSE) of the ensemble model as a function of an increasing number of top-performing models. Source data are provided as a Source Data file[54].

platelet-derived growth factor receptor kinases showed poorer accuracies compared to other kinase families ($P = 0.001$, Kruskal–Wallis test), but these families were better predicted using the Q.E.D and the top-ensemble models (Supplementary Fig. 13). For MAP kinases, the higher prediction error (adjusted $P = 0.016$, Kolmogorov–Smirnov test) could be attributed to the fact that most of the inhibitors targeting MAP kinases are noncompetitive allosteric inhibitors[18]. Similarly, pairs in the CMGC kinase group, including e.g. cyclin-dependent kinases, showed an increased error for bulk of the submissions (adjusted $P = 0.030$, Kolmogorov–Smirnov test), but again both the ensemble and Q.E.D models made better predictions also in this kinase group (Supplementary Fig. 14).

**Comparison against single-dose activity assays**. We next investigated how well the top-performing prediction models compare against the single-dose activity assays in terms of reducing the number of false positives and negatives when selecting most potent compound-kinase activities for more detailed, multi-dose $K_d$ profiling. Such two-step screening approach is widely used in large-scale kinase-profiling studies[5–7,16], where $K_d$ profiling is carried out only for compound-kinase pairs with an inhibition above 80% in the single-dose assays. For this classification task, we defined the ground truth activity classes based on the measured $K_d$ values, which provide a more practical prediction outcome, compared to the rank correlation analyses that already demonstrated predictive

rankings with the top-performing models (Fig. 4). Using the activity cut-off of measured $pK_d = 6$ and a single-dose inhibition cut-off of 80%, similar to previous studies[7,16,19], the positive predictive value (PPV) and the false discovery rate (FDR) of the single-dose assay were PPV = 0.66 and FDR = 0.44, respectively, in the Round 2 dataset. When using the mean aggregation ensemble from the top-performing models and the same cut-off of $pK_d = 6$ for both the predicted and measured activities, we observed an improved precision of PPV = 0.76 and FDR = 0.24.

We repeated the activity classification experiment with multiple $pK_d$ activity cut-offs, and ranked the Round 2 pairs both using the model-predicted $pK_d$ values and the measured single-dose inhibition assay values, and then compared these rankings against the true activity classes based on the measured dose-response assay (with either $pK_d > 6$ or 7 indicating true positive activity). These analyses demonstrated an improved activity classification accuracy using the mean ensemble of the top-performing models (Fig. 7a), especially when focusing on the most potent compound-kinase activities with the highest specificity. This improvement in both sensitivity and specificity was achieved without making any additional activity measurements, and it became even more pronounced with the precision-recall (PR) analysis, which showed that the precision of the ensemble model remained above PPV = 75% level even when the recall (sensitivity) level exceeded 75% (Fig. 7b). The top-performing model (Q.E.D) also showed improved performance when compared to the single-dose activity assay. As expected, the prediction

**Table 1 Model classes, compound and kinase descriptors and training data used by the Round 2 top-performing teams and the baseline model[17].**

| Team | Algorithm type | Algorithm name | Combined models | Training strategy |
| --- | --- | --- | --- | --- |
| DMIS_DK | Deep learning, multi-target learning | Multi-task graph convolutional neural networks | 12 | Train test split |
| AI Winter is Coming | Gradient boosting decision trees | XGboost | 5 per target | K-fold nested cross validation, boosting |
| Q.E.D | Kernel learning | CGKronRLS | 440 | Boosting |
| Gregory Koytiger | Deep learning, artificial neural network | Not applicable | 6 | Fixed hold out |
| Olivier Labayle | Ridge regression | Not applicable | Not applicable | K-fold cross validation |
| Baseline | Kernel learning | CGKronRLS | 1 | K-fold nested cross validation |

| Team | Training data sources | Compound-protein pairs | Bioactivity types | Protein features | Chemical features |
| --- | --- | --- | --- | --- | --- |
| DMIS_DK | DrugTargetCommons, BindingDB | 953521 | $K_d$, $K_i$, $IC_{50}$ | None | Molecular graphs |
| AI Winter is Coming | DrugTargetCommons, ChEMBL | 600000 | $K_d$, $K_i$, $IC_{50}$, $EC_{50}$ %inh, %activity | None | ECFP5, ECFP7, ECFP9, ECFP11 |
| Q.E.D | DrugTargetCommons, ChEMBL, UniProt | 60462 | $K_d$, $K_i$, $EC_{50}$ | Amino acid sequences | ECFP4, ECFP6 |
| Gregory Koytiger | ChEMBL | 250000 | $K_d$, $K_i$, $IC_{50}$ | Amino acid sequences | SMILES strings |
| Olivier Labayle | DrugTargetCommons, ChEMBL, UniProt | 18200 | $K_d$ | K-mer counting | ECFP |
| Baseline | DrugTargetCommons | 44186 | $K_d$ | Amino acid sequences | Path-based fingerprints |

Even if the teams chose to combine predictions from multiple models, they had to submit only one prediction per compound-kinase pair for scoring against the measured activities. Supplementary Table 1 provides further details of all the models submitted together with method surveys and model performances in Round 2.

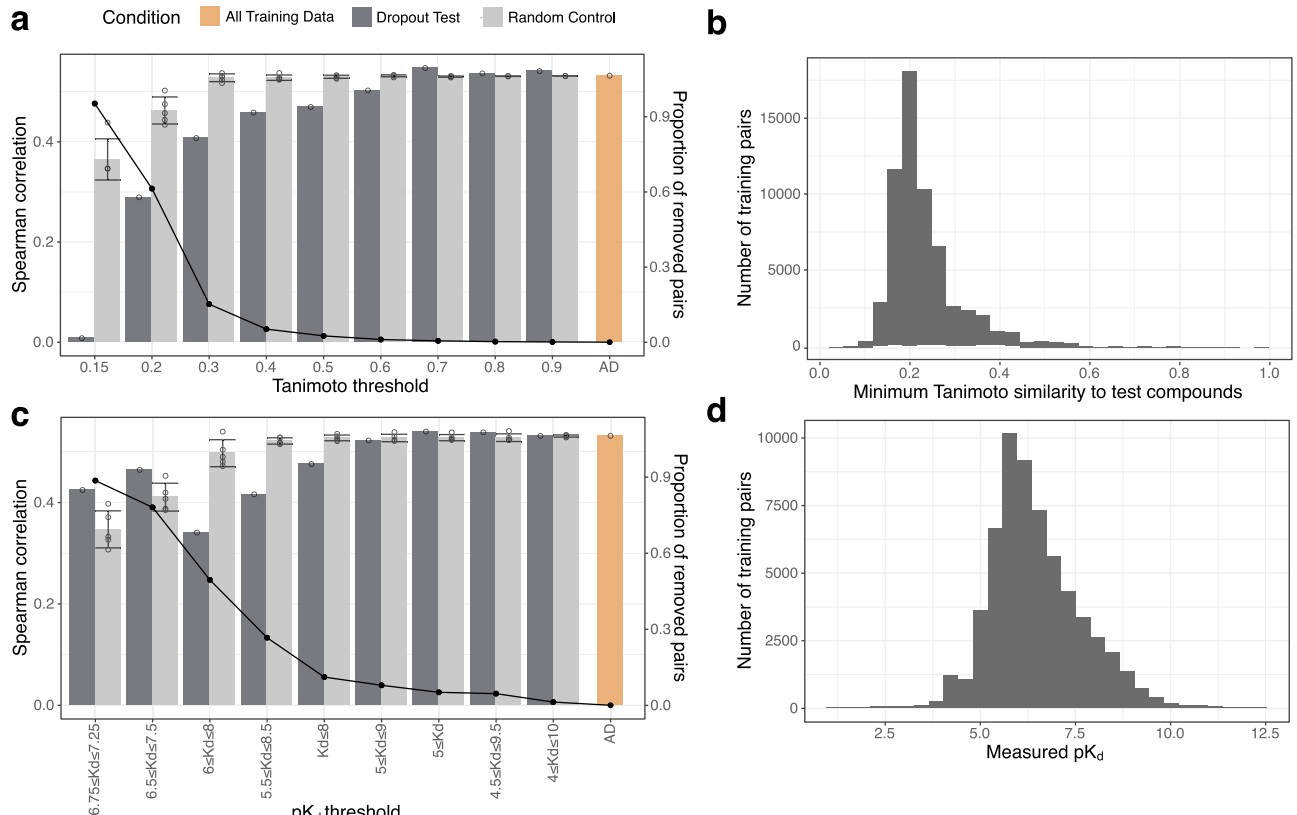

**Fig. 5 The Q.E.D model performance as a function of training data size and scope. a** The drop-out experiment removed increasing numbers of training compounds (as measured by maximum Tanimoto similarity with ECFP4 fingerprint between each training compound and all Round 2 test set compounds), retrained the Q.E.D model, and tested the performance. AD stands for all data. A noticeable decrease in performance begins to appear only at around 0.6 Tanimoto similarity suggesting that highly similar compounds in the training dataset are not necessarily required for accurate model performance. As a control, identical numbers of random compound-kinase pairs were removed, repeated 5 times to assess the variability of random removal. The error bars indicate the standard deviation of these replicates. Black points indicate proportions of removed compound-kinase pairs. **b** A histogram describing the full training dataset used to generate the results in **a**. **c** Model performance with multiple training datasets and varying $pK_d$ levels, where the ranges in the *x*-axis labels refer to the compound-kinase pairs that were included for the model training. AD stands for all data. Random dropout control was repeated 5 times. The error bars indicate the standard deviation of these replicates. **d** A histogram describing the full training dataset used to generate the results in **c**. Source data are provided as a Source Data file[54].

accuracies decreased when using a more stringent measured activity cut-off of $pK_d > 7$ (Supplementary Fig. 15), since these rare extreme activities are more challenging to predict.

**Model-based kinase predictions and their validation**. To further investigate both the sensitivity and specificity of the model predictions, we experimentally profiled 81 additional compound-kinase pairs, which were not part of Round 1 or 2 datasets, selected based on the $pK_d$ predictions from the top-performing models. These post-Challenge experiments were carried out in an unbiased manner, regardless of the compound classes, kinase families, or inhibition levels, to investigate the accuracy of predictive models to identify potent inhibitors of kinases with less than 80% single-dose inhibition; this activity cut-off is often used when selecting pairs for multi-dose $K_d$ testing[7,16,19] but it may miss the more challenging compound-kinase dose-response relationships. Most of the measured $pK_d$ values of these 81 pairs were distributed as expected, according to the expected single-dose inhibition function (Fig. 8a, black trace). However, the model-based approach also identified a large number of unexpected activities ($pK_d > 6$) that had been missed based on the single-dose inhibition assay alone (inhibition <80%); selected examples are discussed below.

As an example of a potent activity missed by the single-dose assay, the ensemble of the top-performing models predicted PYK2 (PTK2B) as a high-affinity target of a PLK inhibitor TPKI-30 (Fig. 8a). The new multi-dose $pK_d$ measurements carried out after the Challenge validated that TPKI-30 indeed has an activity against PYK2 close to its potency towards PLK2 (Fig. 8b, left panel). Neither PYK2 or FAK would have been predicted as potent targets based on the single-dose testing alone, which led to multiple false negatives (Fig. 8b, right panel). In general, the single-dose testing had a relatively low predictivity of the actual TPKI-30 potencies, since kinases other than PLKs with high single-dose activity were confirmed as non-potent targets based on the dose-response $K_d$ testing, resulting in many false positives. In contrast, the top-performing ensemble model predictions turned out to be relatively accurate, except for a few receptor tyrosine kinases (Fig. 8b, left panel). This example shows how the predictive models identify so far unexplored compound-kinase activities missed by standard methods (see also next section).

Another unexpected kinase activity was predicted for GSK1379763 that showed a novel chemotype for inhibition of DDR1 based on the subsequent $K_d$ assays, exceeding that of the AURKB (Fig. 8c, left panel). The single-dose testing suggested that this compound would have potency neither against DDR1 or AURKB (Fig. 8c, right panel), whereas the multi-dose assays

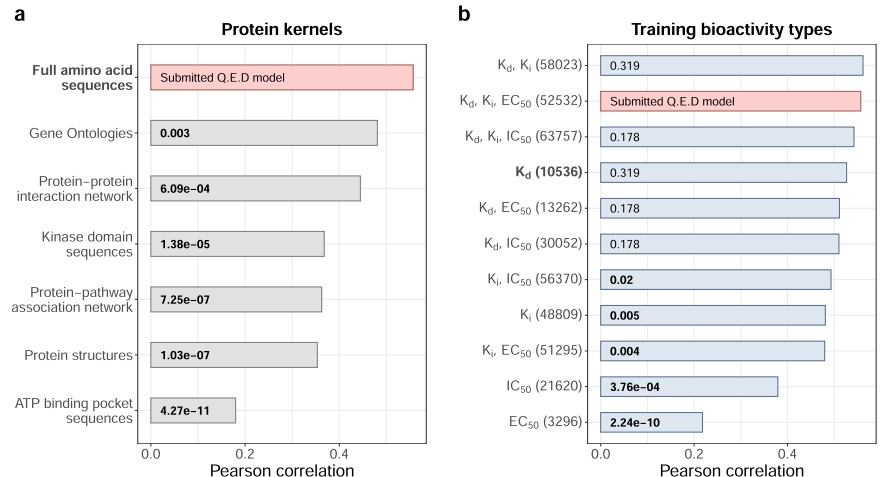

**Fig. 6 The effect of protein descriptors and bioactivity types on Q.E.D model accuracy.** The bars show Pearson correlations between the measured and Q.E.D model-predicted $pK_d$'s calculated over the 394 Round 2 compound-kinase pairs based on different **a** protein kernels and **b** training bioactivity data types. The total number of training bioactivity data points is written in parentheses. The original, submitted Q.E.D model based on the full amino acid sequence-based protein kernel and using $K_d$, $K_i$, and $EC_{50}$ bioactivities in the training dataset is marked with red. No other changes were introduced to the submitted Q.E.D model, which is an ensemble of the regressors with different regularization hyperparameter values and eight compound kernels, but where each regressor is built upon the same protein kernel based on full amino acid sequences. The protein kernel and training bioactivity type used in the baseline model are marked in boldface. The numbers inside the bars are Benjamini–Hochberg adjusted two-sided $P$ values calculated with the Pearson and Filon test for comparing the correlation of the submitted Q.E.D model and each of its re-trained variants. Since the two correlations under comparison are calculated on the same set of data points and they have one variable in common (measured $pK_d$), the dependence between $pK_d$'s predicted by the submitted Q.E.D model and the new model variant is taken into account in the statistical test. Significant $P$ values (adjusted $P < 0.05$) are written in boldface. Source data are provided as a Source Data file[54].

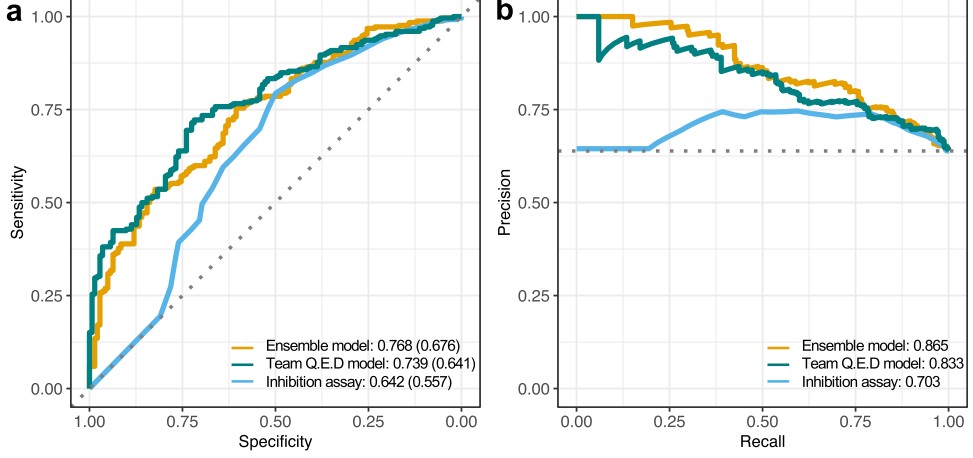

**Fig. 7 Top-performing model predictions compared against single-dose assays. a** Receiver operating characteristic (ROC) curves when ranking the 394 compound-kinase pairs in Round 2 using the $pK_d$ predictions either from the ensemble of the top-performing models (average predicted $pK_d$ from Q.E.D, DMIS_DK and AI Winter is Coming), or only from the Q.E.D model, compared against the experimental single-dose inhibition assays (the pairs with higher inhibition% are ranked first). The true positive activity class contains pairs with measured $pK_d > 6$ (see Supplementary Fig. 15 for $pK_d > 7$). The area under the ROC curve values are shown after the predictors (and the balanced accuracy is marked in the parentheses), and the diagonal dotted line shows the random predictor with an accuracy of AU-ROC = 0.50. **b** Precision-recall (PR) curves for the same activity classification analysis as shown in **a**. The area under the PR curve values are shown after the predictors and the horizontal dotted line indicates the random predictor with a precision of 0.64. Note: Round 2 $K_d$ measurements were pre-selected to include mostly pairs with single-dose inhibition >80%, which makes Round 2 pairs optimal for systematic analysis of false positive predictions, and hence sensitivity (recall) and PPV (precision). However, these 394 pairs pre-selected for $K_d$ profiling were less optimal for a comprehensive analysis of false negative predictions, and the evaluation of specificity. Source data are provided as a Source Data file[54].

confirmed potency towards DDR1 at a similar level as the Round 2 highest affinity target MEK5 (MAP2K5). A novel activity was predicted also between PFE-PKIS14 and CSNK2A2, a dark kinase nominated by the IDG consortium, which was missed by the Round 2 single-dose assay (inhibition = 78%; Fig. 8d, right panel). The single-dose assay led also to a number of other false positive and false negative activities for PFE-PKIS14, whereas the ensemble model demonstrated again a good predictive accuracy

(Fig. 8d, left panel). Arguably, however, this interaction and the ensemble-predicted activity between AKI00000050a and FLT1 could have been identified based on their relatively high single-dose activity, even if less than 80% (Fig. 8a).

**Comparison with other target prediction methods.** To study whether standard target prediction methods could identify the

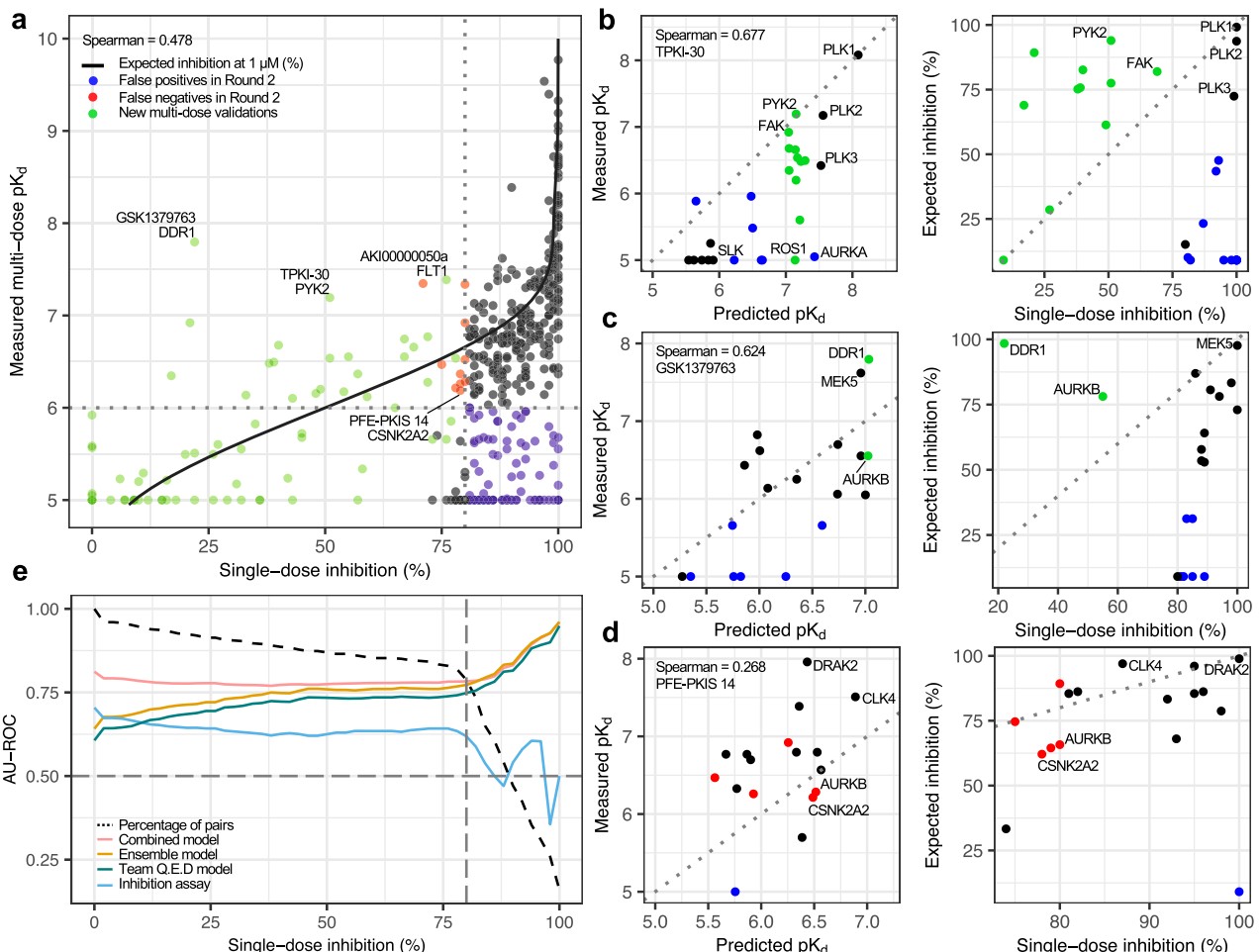

**Fig. 8 Machine learning-based kinase activity predictions. a** Comparison of single-dose inhibition assay (at 1 μM) against multi-dose $K_d$ assay activities across 475 compound-target pairs (395 from Round 2 and 81 from the post-Challenge experiments). The red points indicate false negatives and blue points false positives when using the cut-offs of $pK_d = 6$ and inhibition = 80% among the 394 Round 2 pairs (including 75 pairs with inhibition >80% but that showed no activity in the dose-response assays, i.e, $pK_d = 5$). The green points indicate the new 81 pairs profiled post-Challenge solely based on the ensemble model predictions, regardless of their inhibition levels. The black trace is the expected %inhibition rate based on measured $pK_d$'s, estimated using the maximum ligand concentration of 1 μM both for the single-dose and dose-response assays (see Methods). **b–d** Multi-dose (left) and single-dose (right) assays for kinases tested with TPKI-30, GSK1379763, and PFE-PKIS14. Green points indicate the new experimental validations based on the ensemble model predictions, whereas black points come from Round 2 data. Blue points indicate false positive predictions based either on predictive models or single-dose testing. **e** Predictive accuracy of the top-performing ensemble model (average predicted $pK_d$), top-performing Q.E.D model and single-dose assay (at 1 μM), when classifying subsets of the 475 pairs into the true activity classes with measured $pK_d$ less or higher than 6. The y-axis indicates the area under the receiver operating characteristic (ROC) curve (AU-ROC) as a function of the single-dose inhibition% levels, x-axis the pairs with inhibition >x%, and the dashed black curve the percentage of all pairs that passed that single-dose activity threshold. The combined model trace corresponds to the average of measured and expected inhibition values, where the latter was calculated based on the mean ensemble of the top-performing model $pK_d$ predictions (Q.E.D, DMIS_DK and AI Winter is Coming). See Supplementary Fig. 16 for the corresponding analysis with precision-recall (PR) metric, and Supplementary Fig. 17 for the ROC and PR curves for all the 475 pairs. Source data are provided as a Source Data file[54].

selected compound-target activities predicted by the top-performing ensemble model (Fig. 8), we used the similarity ensemble approach (SEA), a popular target classification method that relates proteins based on chemical similarity among their ligands[20]. Strikingly, the SEA method did not identify target activity among any of the three selected kinases and their confirmed inhibitors (Supplementary Table 2). For instance, the highest scoring hit from SEA for compound TPKI-30 was FAK (PTK2), which belongs to the same subfamily of kinases as PYK2, that was confirmed as potent target of TPKI-30, but their sequence identity is only ~43%. To further model the ligand-receptor interaction between TPKI-30 and PYK2, in the absence of 3D chemical structures, we carried out an in-silico docking procedure. As expected, the protein structure-based docking

approach was not informative enough for predicting the dose-response activity relationships between TPKI-30 and PYK2, but its results supported a potent binding between TPKI-30 and PYK2, with a similar binding affinity compared to the known active ligands that bind to the same binding pocket of PYK2 (Supplementary Fig. 18).

Based on the observation that the single-dose assays and model-based $pK_d$ predictions were overall only weakly correlated (Supplementary Fig. 19), and that they showed opposite trends for the $pK_d$ prediction accuracy when increasing the inhibition cut-off level (Fig. 8e), we finally studied whether the single-dose measurements and the ensemble-based $pK_d$ predictions could be combined for improved kinase activity predictions. Specifically, for each compound-kinase pair, we calculated the average of its

measured and expected inhibition values based on the single-dose assay and ensemble model predictions, respectively. This combined predictor showed improved activity classifications beyond that of the ensemble model predictions, across various inhibition levels, and it identified an extended number of potent compound-kinase interactions at lower single-dose activity, compared to the standard 80% cut-off (Fig. 8d, dotted line). In the full set of all the 475 pairs, the combined model improved both the sensitivity and specificity of the $pK_d$ predictions (Supplementary Fig. 17a), and especially the precision of the top-activity predictions that are prioritized for further validation (Supplementary Fig. 17b). Based on the wider availability of single-dose activity data, this integrated method provides a generally applicable and cost-effective approach for future target activity profiling studies.

## Discussion

While experimental mapping of target activities is critical for understanding compounds' mode of action, biochemical target activity profiling experiments are both time consuming and costly. The enormous size of the chemical universe, spanned by up to $10^{20}$ molecules with potential pharmacological properties[21,22], makes the experimental bioactivity mapping of the full compound and target space quickly infeasible in practice. The IDG-DREAM Drug Kinase Binding Prediction Challenge was designed to benchmark algorithms capable of predicting and prioritizing compound-kinase activities, and therefore to guide data-driven decision making and reduce the high failure rates. The model-guided approach has the potential to help both phenotype-based drug discovery (e.g., mapping of the activity space of lead compounds), and target-based drug discovery (e.g., identification of candidate compounds that selectively inhibit a particular disease-related kinase). As an example, the ensemble of the top-performing models led to a surprising result that the PLK inhibitor TPKI-30 targets also PYK2, and with a somewhat lesser potency also its paralog, FAK (Fig. 8b). Another selected example, CSNK2A2, belongs to the dark kinases nominated by the IDG consortium[23], suggesting that the prediction models can identify potent inhibitors even for the currently understudied kinases. The two other highlighted kinases, PYK2 and DDR1, were neither among the most-studied kinases in terms of the number of dose-response bioactivity data points in the public domain for the model training (Supplementary Fig. 20).

There is an increasing number of studies published each year that introduce new computational algorithms to predict compound-target activities (Supplementary Fig. 21a). Although previous studies have demonstrated the potential of ML algorithms to help fill in the gaps in compound-target interaction maps[17,24], and to accelerate several phases of drug discovery[25,26], to date there has been no systematic and unbiased evaluations of quantitative prediction models for target activity on a blinded and large-enough dataset, such as the one used in the present benchmarking. Participants of this Challenge made use of various ML approaches, which led to relatively wide performance differences (Supplementary Figs. 6 and 7), and covered the most popular ML approaches used for compound-target activity prediction, especially when considering the supervised regression problem (Supplementary Fig. 21b–d; Supplementary Table 1). Only the k-nearest neighbors (kNN) and Bayesian methods were not part of the Challenge submissions. Recently, many advanced deep learning (DL) algorithms have been proposed for compound-target interaction prediction[27–29], and a previous comparative work that used nested cross-validation on bioactivity data from ChEMBL found out that DL methods outperformed other methods, including kNN, support vector machines, random forests, naive Bayes and SEA, as representative target prediction methods[24]. In contrast, our Challenge results did not support the overall superiority of the DL methods compared to the other learning approaches (Fig. 3f).

Among the 31 teams that answered our survey at the end of the Challenge, none of the method classes had a very strong contribution to the prediction accuracy (Supplementary Fig. 22a, b), similarly as has been seen also in other DREAM challenges[30–32]. A striking observation from the survey was that there was a tendency for improved $K_d$ prediction accuracies by teams that used other types of multi-dose bioactivity data (e.g., $K_i$, $IC_{50}$, $EC_{50}$), compared to using $K_d$ data alone (Supplementary Fig. 22c, d). This provides a further opportunity for ML models such as DL that require relatively large training datasets, as these bioactivity types are among the most common in multi-dose target profiling (Supplementary Fig. 22e). Single-dose bioactivity measurements (e.g., potency% and other activity assays) are most abundant in the open bioactivity databases, making their use an exciting option for predicting dose-response activities such as $K_d$. In the Challenge, single-dose %inhibition and %activity data were utilized by one of the top-performing models, AIWIC, whereas the other top performer Q.E.D missed the most abundant multi-dose $IC_{50}$ bioactivities in the model training (Table 1). However, we showed how the integrated use of the other multi-dose bioactivity types, especially $K_i$, compensated for the lack of $IC_{50}$ data and led to the top-performance of the Q.E.D model (Fig. 6b). In contrast, our results based on the Q.E.D model showed that the use of other than kinase proteins and kinase inhibitors in the training data led to a decreased prediction performance compared to the original Q.E.D model with kinases only (Supplementary Fig. 23).

To further study whether the individual models complement each other and could yield an overall better result, we aggregated the top-performing models as a mean ensemble model. Many previous DREAM Challenges have demonstrated that such wisdom of the crowds may improve the predictive power of the individual models through combining models as meta-predictors or ensemble models[30–32]. The ensemble model constructed in this Challenge made use of the various modeling approaches and features of the top-performing models, after which adding more models led to rapid decrease in accuracy (Fig. 4d). In our post-Challenge analyses, the combination of the top-performing ML models improved both the sensitivity and specificity, compared to single-dose target activity assays, without requiring any additional experiments (Fig. 7). We also observed that the combination of the top-performing models using an ensemble model led to accurate and robust predictions of kinase inhibitor potencies across multiple kinase families and groups (Supplementary Figs. 13 and 14). Subsequent target profiling experiments carried out based on the ensemble model predictions demonstrated that the ML models facilitate experimental mapping efforts, both for well-studied and understudied kinases (Fig. 8). Interestingly, combining the single-dose inhibition measurements with the top-performing ML models led to even higher prediction accuracy than using either one alone, while identifying an increased number of potent compound-kinase activities compared to that using the standard 80% inhibition cut-off (Fig. 8e).

The Spearman correlation sub-challenge top performer (Q.E.D) used the same kernel-based regression algorithm as the baseline model[17], yet showed markedly better performance (Fig. 3f). The two models, however, differ in several aspects. The Q.E.D model integrated multiple bioactivity types in their training data, as opposed to using $K_d$ only as was done in the baseline model, and this integrative approach led to significant differences in the prediction accuracy (Supplementary Fig. 12). Although the training dataset sizes of both models had similar numbers of bioactivity values (baseline 44,186 vs. Q.E.D 60,462), Q.E.D used

bioactivity data points for many more compounds than the baseline approach (1968 vs. 13,608 compounds). This increased the diversity of the training dataset, which is often more important than its actual size, especially when majority of the test compounds have no multi-dose bioactivity data available for model training. Furthermore, while both models used the same protein kernel based on Smith–Waterman amino acid sequence alignment, Q.E.D implemented an ensemble model of 440 individual regressors based on various model hyperparameters and eight compound kernels, which resulted in an effective integration of several different compound representations and an improved prediction performance (Supplementary Fig. 24). However, we found that many combinations of the widely used kinase and chemical descriptors led to relatively high prediction accuracies (Fig. 6a; Supplementary Fig. 10), which should make the ensemble approach practical for future applications, also beyond kinases. We also observed that full amino acid sequences used as protein kernels performed significantly better than those based on kinase domain sequences (Fig. 6a). This observation is most likely due to a number of missing kinase domain sequences in the Q.E.D model, which resulted in several $pK_d$ predictions of zero (7%), and reduced training dataset size.

Rather surprisingly, the number of training bioactivity data did not strongly contribute to the prediction accuracies of the top-performing Q.E.D model (Supplementary Fig. 25), provided the training data had sufficient structural diversity for the kinase families being predicted (Fig. 5a). Our training data drop-out analyses have substantial implications for the application of supervised ML in predicting the activity of kinase inhibitors, as they demonstrated that the predictions are reasonably robust even when only limited numbers of structurally similar training data exist (Fig. 5). This observation is also evident from the fact that the top-performing models used a rather different number of training bioactivity values from different multi-dose assays when predicting the $pK_d$ profiles (Table 1). This suggests that the number of training data is not the strongest factor for the predictive performance, rather the way the model is constructed has a much larger contribution to the prediction accuracy, which has implications especially for so-far understudied kinases. Given that the currently available bioactivity data are still rather limited and come in various types, it was comforting to note that the top-performing models made use of the various data types in the training phase (Table 1). This can be considered as another form of 'wisdom of the crowds', and suggests that beyond the community effort for target activity predictions, there is a need for also crowdsourced collection, annotation, and harmonization of different types of bioactivity data to further improve the accuracy and coverage of the predictive models.

To enable the community to apply the predictive models benchmarked in the Challenge to various drug development applications, we have made available the top-performing models as containerized source code. The Docker models enable continuous validation of the model predictions whenever new experimental kinase-profiling data will become available, as well as make it possible to run the best performing models on private data that would otherwise remain closed and unavailable to the research community[33]. The current test data covers ca. 57% of the human protein kinome, and future screening efforts are warranted to extend it to additional interactions with remaining kinases and other important target families. Future applications should select the model class that best fits the specific needs. All the top-performing teams used ML models that leverage information extracted from similar kinases and/or inhibitors to predict the activity of so far unexplored interactions (see Table 1 and Supplementary Table 1). Most of the top-performing models also used amino acid sequences or K-mer counting as target-based

features in their class-agnostic prediction models, and two of the top performers did not utilize any type of protein features. Furthermore, none of the top-models required 3D or other detailed chemical information, making the ML models straightforward to apply for various compound classes. We therefore believe the Challenge models and the current benchmarking results will provide useful information for constructing predictive models also beyond kinases inhibitors.

In conclusion, we envision that the IDG-DREAM Challenge will provide a continuously updated resource for the chemical biology community to benchmark, prioritize, and experimentally test new kinase activities toward accelerating many drug discovery and repurposing applications.

## Methods

**Challenge infrastructure and timeline**. The Challenge was organized and run on the collaborative science platform Synapse. All prediction files were submitted using the Challenge feature of this platform to track which teams and individuals submitted files, and to track the number of submissions per team. Challenge infrastructure scripts including code for calculating the scoring metrics are available at https://github.com/Sage-Bionetworks/IDG-DREAM-Drug-Kinase-Challenge and archived at https://doi.org/10.5281/zenodo.4648011. Teams were permitted to submit three predictions for Round 1, and two predictions for Round 2 (Supplementary Fig. 3). In Round 2, we selected the best of the two submissions for each scoring metric. This led to a selection of 54 final prediction sets for each of the Round 2 scoring metrics (Spearman correlation and RMSE, see 'Scoring of the model predictions' below) from the 99 total submissions in Round 2. For Rounds 1 and 2, we used a common workflow language-based challenge infrastructure to perform the following tasks: (1) validate a prediction file to ensure that it conformed to the correct file structure and had numeric $pK_d$ predictions and return an error email to participants if invalid, (2) run a python script to calculate the performance metrics for a submitted prediction, and (3) return the score to the Synapse platform. For Round 1b, in which we permitted 1 submission a day for 60 days, we implemented a modified Ladderboot[34] protocol to prevent model overfitting. This was done by modifying step (2) above as follows: the scoring infrastructure receive a submitted prediction, check for a previous submission from the same team and run an R script to bootstrap the current and previous submission 10,000 times, calculate a Bayes factor (K) between the two submissions; the scoring harness would then only return an updated score if it was substantially better (K > 3) than the previous submission.

**New bioactivity data for model testing**. To generate unpublished test bioactivity data for scoring of predictions, we sent kinase inhibitors to DiscoverX (Eurofins Corporation) for the generation of new dose-response dissociation constant ($K_d$) values, as a measure of a binding affinity. In order to give a better sense of the relative compound potencies, $K_d$ is represented in the logarithmic scale, as $pK_d = -\log_{10}(K_d)$, where $K_d$ is given in molars [M]. The higher the $pK_d$ value, the higher the inhibitory ability of a compound against a protein kinase. A two-step screening approach was adopted[5–7], where the dose-response $K_d$ values were generated for a range of compound-kinase pairs that had inhibition >80% in the primary single-dose screen using the DiscoverX KINOMEscan protocol (https://www.discoverx.com/services/drug-discovery-development-services/kinase-profiling/kinomescan). KINOMEscan employs a competitive binding assay to estimate $K_d$, wherein the immobilized ligands and the test compound compete for the same binding pocket of the assayed kinase. The compounds were supplied as 10 mM stocks in DMSO, and the top screening concentration was 10 μM in the graded-dose profiling (with one technical replicate). The single-dose assays used a single fixed concentration of 1 μM (no replicates).

A total of 25 of the axitinib-kinase pairs generated for Round 2 were already profiled in previous published studies[7,16], and were therefore excluded from the Round 2 test dataset. The Spearman correlation between these newly measured $pK_d$'s and those available from DTC was 0.701 (Supplementary Fig. 26a), providing the experimental consistency of the $K_d$ measurements for axitinib. We note this 25 $pK_d$'s is a rather limited set for such analysis of consistency, and therefore we extracted a larger set of 416 $K_d$ measurements that overlapped with the Round 2 kinases from two comprehensive target profiling studies[5,6], including 104 pairs where $pK_d = 5$ in both of the studies. The Spearman correlation of these replicate $pK_d$ measurements was 0.842 (Supplementary Fig. 26b), demonstrating a relatively good reproducibility for the large-scale binding affinity measurements. These replicate measurements were also used for determining a practical upper limit of the predictive accuracy of machine learning models in the scoring of their predictions (see below).

The selected kinase targets are a part of the SGC-UNC screening initiative, the Kinase Chemogenomic Set[16]. The primary selection criterion was to investigate the readily screenable human kinome, i.e., kinases with a robust assay readily available through commercial vendors. An additional focus point was to include those screenable kinase targets that are either understudied and/or targets with a Gene

Ontology information available but lacking associated small-molecule activities in ChEMBL[11], called as dark kinases (Tdark) and Tbio targets, respectively[13]. Out of the 392 wild-type human kinases subjected to the screening study by the KGCS Consortium, a subset of 295 kinases were used in our IDG-DREAM Challenge during the Rounds 1 and 2. The 95 kinase inhibitors used in the Challenge (70 for Round 1 and 25 for Round 2) were a part of the kinase inhibitor collection at the SGC-UNC for which we already had the single-dose inhibition screening done at DiscoverX across their large kinase panel (scanMax$^{SM}$).

To subsequently test the top-performing model predictions in additional compound-kinase pairs that were not part of Round 1 or 2 datasets, we selected a set of 88 pairs that showed most potency based on the average predicted pK$_d$ of the top-performing models (Q.E.D, DMIS-DK, and AIWIC), regardless of their single-dose inhibition levels. These 88 pairs were actually scattered across the whole spectrum of single-dose inhibition levels, ranging from 0 to 78% (Supplementary Fig. 19; note: pairs with inhibition >80% were K$_d$-profiled already in Round 2). One of the compounds (TPKI-35) was not available from IDG, so the predicted 7 kinase targets for that compound could not be tested experimentally, resulting in a dataset of total of 81 compound-kinase pairs that were shipped to DiscoverX for multi-dose K$_d$ profiling. One of the compounds (GW819776) was shipped separately in a tube, whereas the other 14 compounds were supplied as 10 μM stocks in DMSO, and the K$_d$ profiling was done using the same KINOMEscan competitive binding assay protocol as for the Round 1 and Round 2 pairs.

**Estimating the expected inhibition levels**. The KINOMEscan assay protocol utilized for both the single-dose and dose-response assays is based on competitive binding assays, where the maximum compound concentration tested was 1 μM and 10 μM respectively. For a given compound-kinase pair, the K$_d$ values calculated from the dose-response assay (excluding pairs with activity ≥10 μM) were then used to estimate the expected single-dose %inhibition level (at 1 μM of compound) using the conventional ligand occupancy formula:

$$\text{Ligand occupancy}(\%) = \frac{\text{Maximum ligand concentration(M)}}{\text{Maximum ligand concentration(M)} + \text{Measured}K_d(M)} \quad (1)$$

In Eq. (1), the maximum ligand concentration is 1 μM in the kinase assay. Therefore, a measured pK$_d$ = 3 (i.e. K$_d$ = $10^{-3}$ M) results in the expected inhibition of 0%, pK$_d$ = 4 and 5 in 1% and 10% expected inhibitions, respectively, and pK$_d$ = 9 (i.e. K$_d$ = $10^{-9}$ M) results in expected inhibition of 100%. The single-dose %inhibition assays were not optimized to accurately estimate the activity values of any specific compound-kinase interaction, leading to a variability in Fig. 8.

**Scoring of the model predictions**. In the Challenge phase, we used the following six metrics to score the quantitative pK$_d$ predictions from the participants:

- Root mean square error (RMSE): square root of the average squared difference between the predicted pK$_d$ and measured pK$_d$, to score continuous activity predictions.
- Pearson correlation: Pearson correlation coefficient between the predicted and measured pK$_d$'s, which quantifies the linear relationship between the activity values.
- Spearman correlation: Spearman's rank correlation coefficient between the predicted and measured pK$_d$'s, which quantifies the ability to rank pairs in correct order.
- Concordance index (CI)[35]: probability that the predictions for two randomly drawn compound-kinase pairs with different pK$_d$ values are in the correct order based on measured pK$_d$ values.
- F1 score: the harmonic mean of the precision and recall metrics. Interactions were binarized by their measured pK$_d$ values into true positive class (pK$_d$ > 7) and true negative class (pK$_d$ ≤ 7).
- Average area under the curve (AUC): average area under ten receiver operating characteristic (ROC) curves generated using ten interaction thresholds based on the measured pK$_d$ interval [6, 8] to binarize pK$_d$'s into true class labels.

The submissions in Round 1 were scored across the six metrics but the teams remained unranked. The Round 2 consisted of two sub-challenges, the top performers of which were determined based on RMSE and Spearman correlation, respectively. Spearman correlation evaluated the predictions in terms of accuracy at ranking the compound-kinase pairs according to the measured K$_d$ values, whereas RMSE considers the AEs in the quantitative binding affinity predictions. The tie-breaking metric for both Rounds was the averaged AUC metric in the ROC analyses that evaluated the accuracy of the models to classify the pK$_d$ values into active and inactive classes based on multiple K$_d$ cutoffs.

In the post-Challenge activity classification analyses, we used two additional metrics that take into account potentially unbalanced class distributions (see also Activity classification analyses):

- PR: area under the PR curve, where precision (PPV) is the fraction of true actives among positive predictions and recall equals to sensitivity.
- Balanced accuracy: the arithmetic mean of the precision and recall metrics. Interactions were binarized into true active class and true inactive class based on the measured pK$_d$ values.

Two different activity cut-offs were applied (measured pK$_d$ > 6 or 7) to study how the ground truth class balance affects the results (see Fig. 7, and Supplementary Figs. 15–17). The same cut-off value was used for the predicted pK$_d$ to calculate the balanced accuracy.

**Statistical evaluation of the predictions**. Determination of the top performers was made by calculation of a Bayes factor relative to the top-ranked submission in each category. Briefly, we bootstrapped all submissions (10,000 iterations of sampling with replacement), and calculated RMSE and Spearman correlation to the test dataset to generate a distribution of scores for each submission. A Bayes factor was then calculated using the challengescoring R package (https://github.com/sage-bionetworks/challengescoring) for each submission relative to the top submission in each sub-challenge. Submissions with a Bayes factor K ≤ 3 relative to the top submission were considered to be tied as top performers. Tie breaking for both sub-challenges was performed by identifying submission with the highest average AUC. To create a distribution of random predictions, we randomly shuffled the 430/394 K$_d$ values across the set of 430/394 compound-kinase pairs in the Round 1/ Round 2 datasets, and repeated the permutation procedure 10,000 times. Then we compared the actual Round 1/Round 2 prediction scores to Spearman and RMSE calculated from the permuted K$_d$ data. We defined a prediction as better than random if its score was higher than the maximum of the 10,000 random predictions (empirical P = 0.0, non-parametric permutation test).

Statistical comparison of the predictions in terms of the two winning metrics was performed using either two-sample or paired Wilcoxon tests (non-parametric tests), depending whether groups of participants or the same participants were compared between the two Challenge scoring rounds. We compared the magnitudes of Pearson correlations between the measured and predicted pK$_d$'s from two different models using Pearson and Filon test for two overlapping correlations implemented in cocor[36] R package. Specifically, since the two correlations under comparison were calculated on the same set of compound-kinase pairs and have one variable in common (measured pK$_d$), the correlation between pK$_d$'s predicted by two different models is taken into account in the statistical test. Parametric test was applied in these comparisons due to the large number of compound-target pairs in Round 2 (394 pairs). When analysing the questionnaire's results, statistical significance was assessed using the non-parametric Kruskal–Wallis test, adjusted for multiple comparisons with Benjamini–Hochberg control of FDR. All the measurements corresponded to distinct participants or teams in the Challenge.

To determine the maximum possible performance practically achievable by any computational models, we utilized replicate K$_d$ measurements from distinct studies that applied a similar biochemical assay protocol. We used the DrugTargetCommons to retrieve 863 and 835 replicated K$_d$ values for kinases or compounds that overlapped with the Round 1 and 2 datasets, respectively. These data originated from two comprehensive screening studies[5,6]. To better represent the distribution of pK$_d$ values in the test data, we subset the DTC data to contain 35% (Round 1) and 25% (Round 2) pK$_d$ = 5 values, approximately matching the proportion of pK$_d$ = 5 values in Round 1 and Round 2 test sets. For Round 1, we used 317 replicated K$_d$'s, including 111 randomly selected pairs where pK$_d$ = 5. For Round 2, we used 416 replicated K$_d$'s, including 104 randomly selected pairs where pK$_d$ = 5. We randomly sampled the replicate measurements of these compound-kinase pairs (with replacement), calculated the Spearman correlation and RMSE between the pK$_d$'s of the two studies for each 430 and 394 sub-sampled sets for Round 1 and 2, respectively, and repeated this procedure for a total of 10,000 samplings.

**The baseline prediction model**. We used a recently published and experimentally validated kernel regression framework as a baseline model for compound-kinase binding affinity prediction[17]. Our training dataset consisted of 44,186 pK$_d$ values (between 1968 compounds and 423 human kinases) extracted from DTC. Median was taken if multiple pK$_d$ measurements were available for the same compound-kinase pair. We constructed protein kinase kernel using normalized Smith–Waterman alignment scores between full amino acid sequences, and four Tanimoto compound kernels based on the following fingerprints implemented in rcdk R package[37]: (i) 881-bit fingerprint defined by PubChem (pubchem), (ii) path-based 1024-bit fingerprint (standard), (iii) 1024-bit fingerprint based on the shortest paths between atoms taking into account ring systems and charges (shortestpath), and (iv) extended connectivity 1024-bit fingerprint with a maximum diameter set to 6 (ECFP6; circular). We used CGKronRLS as a learning algorithm (implementation available at https://github.com/aatapa/RLScore)[38]. We conducted a nested cross-validation in order to evaluate the generalization performance of CGKronRLS with each pair of kinase and compound kernels as well as to tune the regularization hyperparameter of the model. In particular, since the majority of the compounds from the Challenge test datasets had no bioactivity data available in the public domain, we implemented a nested leave-compound-out cross-validation to resemble the setting of the Challenge as closely as possible. The model comprising protein kernel coupled with compound kernel built upon path-based fingerprint (standard) achieved the highest predictive performance on the training dataset (as measured by RMSE), and therefore it was used as a baseline model for compound-kinase binding affinity prediction in both Challenge Rounds.

**Top-performing models**. Supplementary write ups provide details of all qualified models submitted to the Challenge[39]. The key components of the top-performing models are listed in Table 1 and summarized below.

*Team Q.E.D model.* To enable a fine-grained discrimination of binding affinities between similar targets (e.g., kinase family members), the team Q.E.D explicitly introduced similarity matrices of compounds and targets as input features into their regression model. The regression model was implemented as an ensemble version (uniformly averaged predictor) of 440 CGKronRLS regressors (CGKronRLS v0.81)[38,40], but with different choices of regularization strengths [0.1, 0.5, 1.0, 1.5, 2.0], training epochs [400, 410, …, 500], and similarity matrices: the protein similarity matrix was derived based on the normalized striped Smith–Waterman alignment scores[41] between full protein sequences (https://github.com/mengyao/Complete-Striped-Smith-Waterman-Library). Eight different alternatives of compound similarity matrices were computed using both Tanimoto and Dice similarity metrics for different variants of 1024-bit Morgan fingerprints[42] ('radius' [2, 3] and 'useChirality' [True, False], implementation available at https://github.com/rdkit/rdkit). Unlike the baseline method, which used only the available $pK_d$ values from DTC for training, the team Q.E.D model extracted 16,945 $pK_d$, 53,894 $pK_i$, and 3301 $pEC_{50}$ values from DTC. After merging the same compound-kinase pairs from different studies by computing their medians, 60,462 affinity values between 13,608 compounds and 527 kinases were used as the training data.

*Team DMIS_DK model.* Team DMIS_DK built a multi-task Graph Convolutional Network (GCN) model based on 953,521 bioactivity values between 474,875 compounds and 1474 proteins extracted from DTC and BindingDB. Three types of bioactivities were considered, that is, $pK_d$, $pK_i$, and $pIC_{50}$. Median was computed if multiple bioactivities were present for the same compound-protein pair. Multi-task GCN model was designed to take compound SMILES strings as an input, which were then converted to molecular graphs using RDKit python library (http://www.rdkit.org). Each node (i.e. atom) in a molecular graph was represented by a 78-dimensional feature vector, including the information of atom symbol, implicit valence, aromaticity, number of bonded neighbors in the graph, and hydrogen count. No protein descriptors were utilized. The final model was an ensemble of four multi-task GCN architectures described in the Supplementary writeups[39]. For the Challenge submission, the binding affinity predictions from the last K epochs were averaged, and then the average was taken over the 12 multi-task GCN models (four different architectures with three different weight initializations). Hyperparameters of the multi-task GCN models were selected based on the performance on a hold-out set extracted from the training data. The GCN models were implemented using PyTorch Geometric (PyG) library[43].

*Team AI Winter is Coming model.* Team AI Winter is Coming built their prediction model using Gradient Boosted Decision Trees (GBDT) implemented in XGBoost algorithm (xgboost v0.90, scikit-learn v0.20.3)[44]. Training dataset included 600,000 $pK_d$, $pK_i$, $pIC_{50}$, and $pEC_{50}$ values extracted from DTC and ChEMBL (version 25), considering only compound-protein pairs with ChEMBL confidence score of 6 or greater for 'binding' or 'functional' human kinase protein assays. For a given protein target, replicate compounds with different bioactivities in a given assay (differences larger than one unit on a log scale) were excluded. For similar replicate measurements, a single representative assay value was selected for inclusion in the training dataset. Chemical data was standardized using the ChemAxon Standardizer v18 (https://www.chemaxon.com), and further processed with OpenEye chemistry toolkit (Software Inc, https://www.eyesopen.com/oechem-tk). Each compound was characterized by a 16,000-dimensional feature vector being a concatenation of four ECFP fingerprints (as implemented in RDKit) with a length set to 5, 7, 9, and 11. No protein descriptors were used in the XGBoost algorithm[44]. A separate model for each protein target was trained using nested cross-validation (CV), where inner loops were used to perform hyperparameter optimization and recursive feature elimination. The final binding affinity prediction was calculated as an average of the predictions from the cross-validated models based on five outer CV loops.

**Training data dropout experiments**. We developed Docker containers using the Team Q.E.D model that accepted input parameters for minimum Tanimoto similarity to the test dataset (similarity calculated using the ECFP4 fingerprint), or $pK_d$ cutoff values, to eliminate training data based on various thresholds (see Data and Code Availability). For each condition, training data were dropped out, the model was trained on the remaining data, and the trained model generated predictions for the Round 2 test compound-kinase $pK_d$ values. The predicted $pK_d$ values for each training condition were then scored by calculating the Spearman correlation in the test dataset. We trained and tested each experimental condition once. As a control for each experimental condition, we randomly removed an equivalent number of training compounds, repeated 5 times per condition.

**Ensemble model construction**. Ensemble models were generated by combining the best-scoring Round 2 predictions from each team. We iteratively combined models starting from the highest scoring Round 2 prediction (e.g., ensemble #1—highest scoring prediction, ensemble #2—second highest scoring,

ensemble #3—third highest scoring, and so on) for all 54 Round 2 submitting teams. Three types of ensembles were created using arithmetic mean, median, and rank-weighted summarization approaches. The rank-weighted ensemble was calculated by multiplying each set of predictions by the total number of submissions plus 1 minus the rank of the prediction file, summing these weighted predictions, and then dividing by the sum of the multiplication factors. The 54 ensemble predictions for each of the three summary metrics were bootstrapped and Bayes factors were calculated as described in the 'Statistical evaluation of the predictions' Methods section to determine which models were substantially different from the top-ranked submission. We also randomly sampled 1000 sets of 4 models among the Challenge submissions, ensembled the predictions in each set, and scored each set. These combinations of four random-performance models could not match or supersede the performance of an ensemble of the top four models (i.e., an empirical $P = 0.0$, Supplementary Fig. 8).

**Activity classification analyses**. To compare the top-performing prediction models and their ensemble against the single-dose activity assay, the standard confusion matrix was constructed using the measured $pK_d$ values to define the true positive and true negative classes for the 394 pairs in Round 2, using either $pK_d > 6$ or $pK_d > 7$ for indicating true positive activity. The predicted positive and negative classes for the pairs were defined based on either the single-dose activity measurement, using inhibition cut-off of 80%[7,16,19], or the model-predicted $pK_d$ values, using the same activity thresholds as with the measured $pK_d$ values (i.e., either $pK_d = 6$ or $pK_d = 7$). PPV and FDR were calculated as the classification performance scores. The lower threshold of measured $pK_d = 6$ was used in the classification evaluations to have more balanced true positive and negative classes. To carry out a more systematic analysis of the model prediction accuracies, the 394 pairs in Round 2 were ranked both using the model-predicted $pK_d$ values and the measured single-dose %inhibition values, and then these rankings were compared against the ground-truth activity classification based on the dose-response measurements (using again either $pK_d > 6$ or $pK_d > 7$ for indicating the true positive activity). The results were visualized using both ROC and PR curves, implemented in the pROC and pRROC R-packages, respectively[45,46]. The area under the ROC curve (AU-ROC) and PR curve (PR-AUC) were calculated as summary classification performance metrics.

**Class enrichment analyses**. For each of the 394 compound-kinase pairs from the Round 2 test set, we calculated an AE (i.e., residual errors between predicted and measured $pK_d$ values) considering (i) 90 out of all 99 submissions with average AE below 2, (ii) Spearman correlation-based mean aggregation ensemble model, and (iii) the best submission from the top-performing Q.E.D team. We computed median AE across 90 submissions and, in each case (i–iii), we ranked all the compound-kinase pairs according to their AE (from highest to lowest AE). To explore whether any of the pre-defined kinase classes were enriched among the predictions with the highest or lowest AE, we applied the enrichment analysis implemented in the clusterProfiler R package[47]. In this tool, the enrichment $P$ values were calculated based on a weighted Kolmogorov–Smirnov-like statistic, similar to gene set enrichment analysis (GSEA). We considered the classes defined based on kinase families and kinase groups.

**PubMed literature scan**. A total of 959 abstracts of drug-target interaction prediction publications were extracted from PubMed (on 16 February 2021) using easyPubMed R package[48] with the following query: (("compound target") OR ("target affinity") OR ("drug target") OR ("binding affinity")) AND (("prediction") OR ("algorithm")) AND ("computational") NOT (review[Publication Type]) NOT (news[Publication Type]) NOT (newspaper article[Publication Type]) NOT (systematic review[Publication Type]) NOT (editorial[Publication Type]). textmineR[49] and SnowballC[50] R packages were used to convert all words in the abstracts to lowercase, remove punctuation, numbers and stop words, as well as perform stemming. Next, 4847 n-grams of size up to three and occurring in at least five abstracts were extracted and manually curated to keep only n-grams related to machine learning methods (e.g., deep_neural, deep_learn, kernel_base) and problem classes (e.g., classif_model, regress_model, supervis_learn). Finally, the resulting n-grams were grouped (e.g., both deep_neural and deep_learn bigrams represent deep learning methods), and the various modeling approaches used by the Challenge teams were mapped into the approaches based on the literature scan. A co-occurrence graph of the problem classes and machine learning methods was created using the igraph[51] R package.

**Existing target prediction methods**. We applied the online SEA web-application (http://sea.bkslab.org/search) to make target predictions for the three compounds highlighted in the revised manuscript, TPKI-30, GSK1379763 and PFE-PKIS14, for which Q.E.D model-predicted strong activity against DDR1, PYK2 (PTK2B) and CSNK2A2 ($pK_d > 6$), and which were experimentally validated post-Challenge. In the SEA method, we used the ECFP4 fingerprints that were also used by the top-performing prediction models in the Challenge (see Table 1).

To model the interaction between TPKI-30 and PYK2 (PDB entry 5TO8 [https://doi.org/10.2210/pdb5TO8/pdb]), we carried out binding affinity predictions of various active ligands with docking study in terms of their measured

$pK_d/pK_i$ activity values. The docking was done with AutoDock Vina[52]. The X-ray crystal structure of protein PYK2 (PDB entry 5TO8 [https://doi.org/10.2210/pdb5TO8/pdb]) was obtained from RCSB[53], and a collection of 26 compounds (including TPKI-30), with potent activity towards PYK2 (i.e., $pK_d/pK_i > 6$) from ChEMBL[11], BindingDB[12], and DTC[14], were used as ligands in the docking procedure.

**Reporting summary**. Further information on research design is available in the Nature Research Reporting Summary linked to this article.

## Data availability

Challenge Round 1 and Round 2 $pK_d$ datasets are available from DrugTargetCommons (https://drugtargetcommons.fimm.fi/), and in Supplementary Data 1. The $pK_d$ values of additional compound-target pairs selected for post-Challenge DiscoverX profiling are available in Supplementary Data 2. Source data underlying the figures and display items are provided on Zenodo[54] (subdirectory source_data) and with this paper as a Source Data file. The study made use of the following publicly available databases: Druggable Genome (IDG) consortium (https://druggablegenome.net/), ChEMBL (https://www.ebi.ac.uk/chembl/), BindingDB (https://www.bindingdb.org), IDG Pharos (https://pharos.nih.gov/), DrugTargetCommons (https://drugtargetcommons.fimm.fi/), Synapse (https://www.synapse.org/). The crystal structure to model the interaction between TPKI-30 and PYK2 was obtained from the RCSB PDB (https://www.rcsb.org/) with the PDB code 5TO8 [https://doi.org/10.2210/pdb5TO8/pdb]. Source data are provided with this paper.

## Code availability

The Docker containers of the top-performing teams are available on Synapse[55]. Please refer to the Synapse.org documentation (https://docs.synapse.org/articles/docker.html) for guidance on using the Synapse Docker repository. The codes for reproducing the results and figures are available at GitHub (https://github.com/Sage-Bionetworks/IDG-DREAM-Challenge-Analysis/) and archived in Zenodo[54]. Key R packages used beyond those mentioned elsewhere in Methods include tidyverse[56] and the Synapse Python Client (https://github.com/Sage-Bionetworks/synapsePythonClient); packages used and their versions are listed in the renv lockfile in the Github and Zenodo repositories.

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

## Acknowledgements

The authors thank the IDG Kinase Data and Resource Generation Center for generating new sets of target activity data for the Challenge Rounds 1 and 2, Olle Hansson (FIMM) for technical assistance with DrugTargetCommons platform, Tianduanyi Wang (FIMM) for his help with the baseline submissions, Anna Goldenberg (University of Toronto, Canada) and Chloe-Agathe Azencott (Institut Curie, France) for organizing the DREAM Idea Challenge, and Barbara Rieck and Ladan Naghavian for the bioactivity profiling at DiscoverX (Eurofins Corporation). T.A. acknowledges support from the Academy of Finland (grants 310507, 313267, 326238), Cancer Research UK and the Brain Tumour Charity (grant REF: C42454/A28596), and Helse Sør-Øst (grant No. 2020026). C.W., T.W., D.D. acknowledge support from the National Institutes of Health (1U24DK116204-01). The SGC is a registered charity that receives funds from AbbVie, Bayer Pharma AG, Boehringer Ingelheim, Canada Foundation for Innovation, Eshelman Institute for Innovation, Genome Canada, Innovative Medicines Initiative (ULTRA-DD 115766), Wellcome Trust, Janssen, Merck Kga, Merck Sharp & Dohme, Novartis Pharma AG, Ontario Ministry of Economic Development and Innovation, Pfizer, São Paulo Research Foundation-FAPESP, and Takeda. O.I. acknowledges support from the National Science Foundation (NSF CHE-1802789 and CHE-2041108), and Eshelman Institute for Innovation (EII) awards. O.I. thanks the OpenEye Free Academic Licensing Program for providing a free academic license for their chemistry toolkit. M.P. acknowledges support from The Molecular Sciences Software Institute (MolSSI) Software Fellowship and NVIDIA Graduate Fellowship. We gratefully acknowledge the support and hardware donation from NVIDIA Corporation. J.G. acknowledges support from the National Institutes of Health (U54OD020353). T.I.O. acknowledges support from the National Institutes of Health (U24CA224370; U24TR002278; U01CA239108).

## Author contributions

Conceptualization: A.C., B.R., R.J.A., K.D., A.S., D.H.D., G.S., K.W., J.G., T.A.; data curation: A.C., B.R., R.J.A., A.L., C.I.W., T.M.W., D.H.D.; formal analysis: A.C., B.R., R.J.A., F.W. S.P., O.I., S.L., M.M., Z.T., M.J., S.K., M.P., S.C., J.Z., T.A.; funding acquisition: O.I., M.P., C.W., T.M.W., T.I.O., D.H.D., K.W., J.G., T.A.; investigation: A.C., B.R., R.J.A., C.I.W., D.H.D., T.A.; methodology: A.C., B.R., R.J.A., F.W., S.P., O.I., S.L., M.M., A.L., Z.T., M.J., S.K., M.P., S.C., J.Z., K.D., G.K., J.K., T.A.; project administration: T.I.O. D.H.D., G.S., J.G., T.A.; resources: A.C., B.R., R.J.A., A.L.; software: A.C., B.R., R.J.A., A.L.; supervision: K.W., J.G., T.A.; validation: A.C. B.R., R.J.A., D.H.D., K.W., T.A.; visualization: A.C., B.R., R.J.A., T.A.; writing—original draft: A.C., B.R., R.J.A., F.W., S.P., O.I., S.L. M.M., A.L., Z.T., M.J., S.K., M.P., S.C., J.Z., K.D., G.K., J.K., C.I.W. T.M.W., T.I.O, A.S., D.H.D., G.S., K.W., J.G., T.A.; writing—review and editing: A.C., B.R., R.J.A., O.I., T.I.O., A.S., K.W., J.G., TA.

## Competing interests

The SGC is a registered charity that receives funds from AbbVie, Bayer Pharma AG, Boehringer Ingelheim, Canada Foundation for Innovation, Eshelman Institute for Innovation, Genome Canada, Innovative Medicines Initiative (ULTRA-DD 115766), Wellcome Trust, Janssen, Merck Kga, Merck Sharp & Dohme, Novartis Pharma AG, Ontario Ministry of Economic Development and Innovation, Pfizer, São Paulo Research Foundation-FAPESP, and Takeda. T.I.O. has received honoraria or consulted for Abbott, AstraZeneca, Chiron, Genentech, Infinity Pharmaceuticals, Merz Pharmaceuticals, Merck Darmstadt, Mitsubishi Tanabe, Novartis, Ono Pharmaceuticals, Pfizer, Roche, Sanofi and Wyeth. J.Z. is founder and CTO of Silexon AI Technology Co. Ltd. and has an equity interest. The rest of the authors declare no competing interests.

## Additional information

¹Institute for Molecular Medicine Finland (FIMM), HiLIFE, University of Helsinki, Helsinki, Finland. ²Department of Computer Science, Helsinki Institute for Information Technology (HIIT), Aalto University, Espoo, Finland. ³Department of Computing, University of Turku, Turku, Finland. ⁴Computational Oncology, Sage Bionetworks, Seattle, WA, USA. ⁵Institute for Interdisciplinary Information Sciences, Tsinghua University, Beijing, China. ⁶Department of Computer Science and Engineering, Korea University, Seoul, Republic of Korea. ⁷Department of Chemistry, Carnegie Mellon University, Pittsburgh, PA, USA. ⁸Laboratory for Molecular Modeling, Division of Chemical Biology and Medicinal Chemistry, UNC Eshelman School of Pharmacy, University of North Carolina, Chapel Hill, NC, USA. ⁹Immuneering Corporation, Cambridge, MA, USA. ¹⁰Structural Genomics Consortium, UNC Eshelman School of Pharmacy, University of North Carolina, Chapel Hill, NC, USA. ¹¹Translational Informatics Division and Comprehensive Cancer Center, University of New Mexico School of Medicine, Albuquerque, NM, USA. ¹²Department of Pharmacological Sciences, Icahn School of Medicine at Mount Sinai, New York, NY, USA. ¹³IBM T J Watson Research Center, IBM, Yorktown Heights, NY, USA. ¹⁴Biotech Research and Innovation Centre (BRIC), University of Copenhagen, Copenhagen, Denmark. ¹⁵Department of Mathematics and Statistics, University of Turku, Turku, Finland. ¹⁶Institute for Cancer Research, Oslo University Hospital, Oslo, Norway. ¹⁷Oslo Centre for Biostatistics and Epidemiology (OCBE), University of Oslo, Oslo, Norway. ¹⁸Dept. of Computer Engineering, TOBB University of

Economics and Technology, Ankara, Turkey. [19]Institute of Biomedical Sciences, Academia Sinica, Taipei, Taiwan. [20]Department of Biomedical Engineering, University of California, Davis, CA, USA. [21]Department of Biostatistics, Harvard School of Public Health, Boston, MA, USA. [22]Department of Environmental Health, College of Medicine, University of Cincinnati, Cincinnati, OH, USA. [23]Department of Computer Engineering, VIIT, Pune, India. [24]Applied Artificial Intelligence Institute, Deakin University, Geelong, VIC, Australia. [25]Department of Electrical and Computer Engineering, Texas A&M University, College Station, TX, USA. [26]Microsoft, One Microsoft Way, Redmond, WA, USA. [27]The University of Pennsylvania, Philadelphia, PA, USA. [28]Department of Computer Science and Engineering, Texas A&M University, College Station, TX, USA. [29]The University of Texas at Austin, Austin, TX, USA. [30]Department of Computer Engineering, Bogazici University, Istanbul, Turkey. [31]Department of Chemical Engineering, Bogazici University, Istanbul, Turkey. [32]Department of Biochemistry, Graduate Center, The City University of New York, New York, NY, USA. [33]Department of Computer Science, Hunter College, The City University of New York, New York, NY, USA. [34]Department of Neurosurgery, Cancer Center Amsterdam CCA, De Boelelaan 1117, Amsterdam, The Netherlands. [35]Department of Drug Design and Pharmacology, University of Copenhagen, Copenhagen, Denmark. [36]BioSys Lab, National Technical University of Athens, Athens, Greece. [37]Department of Biotechnology, Ghent University, Ghent, Belgium. [38]Department of Data Analysis and Mathematical Modelling, Ghent University, Ghent, Belgium. [39]Department of Computer Science, University of Illinois at Urbana-Champaign, Urbana, IL, USA. [40]School of Medicine, Tsinghua University, Beijing, China. [41]Department of Computer and AI Engineering, Hacettepe University, Ankara, Turkey. [42]Department of Computer Engineering, METU, Ankara, Turkey. [43]KanSiL, Department of Health Informatics Graduate School of Informatics, METU, Ankara, Turkey. [44]European Molecular Biology Laboratory European Bioinformatics Institute (EMBL-EBI), Hinxton, Cambridge, UK. [45]Semmelweis University, Department of Physiology, Budapest, Hungary. [46]Max-Planck-Institute for Molecular Genetics, Department Computational Molecular Biology, Berlin, Germany. [47]University of Potsdam, Department of Computer Science, Potsdam, Germany. [48]MicroDiscovery GmbH, Berlin, Germany. [49]Division of Electronics, Ruđer Bošković Institute, Zagreb, Croatia. [50]NMR Centre, Ruđer Bošković Institute, Zagreb, Croatia. [51]These authors contributed equally: Anna Cichońska, Balaguru Ravikumar, Robert J. Allaway. [52]These authors jointly supervised this work: Krister Wennerberg, Justin Guinney, Tero Aittokallio. *A list of authors and their affiliations appears at the end of the paper. ✉email: krister.wennerberg@bric.ku.dk; jguinney@gmail.com; tero.aittokallio@helsinki.fi

## The IDG-DREAM Drug-Kinase Binding Prediction Challenge Consortium

**User oselot** Mehmet Tan[18]

**Team N121** Chih-Han Huang[19], Edward S. C. Shih[19], Tsai-Min Chen[19], Chih-Hsun Wu[19], Wei-Quan Fang[19], Jhih-Yu Chen[19] & Ming-Jing Hwang[19]

**Team Let_Data_Talk** Xiaokang Wang[20], Marouen Ben Guebila[21], Behrouz Shamsaei[22] & Sourav Singh[23]

**User thinng** Thin Nguyen[24]

**Team KKT** Mostafa Karimi[25,26], Di Wu[25,27], Zhangyang Wang[28,29] & Yang Shen[25]

**Team Boun** Hakime Öztürk[30], Elif Ozkirimli[31] & Arzucan Özgür[30]

**Team KinaseHunter** Hansaim Lim[32] & Lei Xie[33]

**Team AmsterdamUMC-KU-team** Georgi K. Kanev[34], Albert J. Kooistra[35] & Bart A. Westerman[34]

**Team DruginaseLearning** Panagiotis Terzopoulos[36], Konstantinos Ntagiantas[36], Christos Fotis[36] & Leonidas Alexopoulos[36]

**Team KERMIT-LAB - Ghent University** Dimitri Boeckaerts[37], Michiel Stock[38], Bernard De Baets[38] & Yves Briers[37]

**Team QED** Fangping Wan[5], Shuya Li[5] & Yunan Luo[39], Hailin Hu[40], Jian Peng[39] & Jianyang Zeng[5]

**Team METU_EMBLEBI_CROssBAR** Tunca Dogan[41], Ahmet S. Rifaioglu[42], Heval Atas[43], Rengul Cetin Atalay[43], Volkan Atalay[42] & Maria J. Martin[44]

**Team DMIS_DK** Sungjoon Park[6], Minji Jeon[6], Sunkyu Kim[6] & Junhyun Lee[6], Seongjun Yun[6], Bumsoo Kim[6], Buru Chang[6] & Jaewoo Kang[6]

**Team AI Winter is Coming** Mariya Popova[7], Stephen Capuzzi [8] & Olexandr Isayev [7]

**Team hulab** Gábor Turu[45], Ádám Misák[45], Bence Szalai[45] & László Hunyady[45]

**Team ML-Med** Matthias Lienhard[46], Paul Prasse[47], Ivo Bachmann[48], Julia Ganzlin[47], Gal Barel[46] & Ralf Herwig[46]

**Team Prospectors** Davor Oršolić[49], Bono Lučić[50], Višnja Stepanić[49] & Tomislav Šmuc[49]

**Challenge organizers** Anna Cichońska [1,2,3,51], Balaguru Ravikumar [1,51], Robert J. Allaway [4,51], Michael Mason [4], Andrew Lamb[4], Ziaurrehman Tanoli [1], Kristen Dang[4], Carrow I. Wells [10], Timothy M. Willson [10], Tudor I. Oprea [11], Avner Schlessinger [12], David H. Drewry [10], Gustavo Stolovitzky [13], Krister Wennerberg [14,52✉], Justin Guinney[4,52✉] & Tero Aittokallio [1,2,15,16,17,52✉]

