## [Peer Review File · Nature Communications]

Reviewers' comments:

Reviewer #3 (Remarks to the Author):

The manuscript by Wennerberg, Guinney, Aittokallio and co-workers reports on a competition for implementing machine learning models to predict bioactivity values of small molecules against several kinases. This is relevant research as it might enable the adoption and standardization of certain workflows in this space, while identifying potential pitfalls in model building practices and available chemical/biological data. In spite of this, I did find the message of the manuscript confusing, which made it difficult to fully gauge the significance of the results and methods. Moreover, the authors identify many research questions, but unfortunately, their data is not exceptional at addressing either of them. Overall, the findings in this study (including their validation and advances to state-of-the-art) are publishable, but fall short of what is expected from a Nature Communications manuscript.

Below I provide a series of comments that support my assessment.

- 1) It is confusing how the predictive accuracy of a given model or ensemble of models can exceed that of a kinase activity assay. An activity assay does not predict activity but provides activity data. Whether the obtained value is accurate or not is a different question but, as it stands, the comparison is not correct.
- 2) I understand that regression models were built to predict affinities, yet the authors mention that some of the activities found in this study were unexpected even for under-studied targets. From a medicinal chemistry and even machine learning point of view this does not make sense. If regression models were built, then a certain amount of data is present; thus the targets have been studied to a considerable extent. It would be different if unsupervised learning had been employed, but this is not the case.
- 3) The authors mention that chemical agents inhibiting single protein targets are rare and that a comprehensive understanding of pharmacological effects is important. The study does not seem to answer this need as all models are focused on a subset of kinases, and ignoring several potentially important interactions with proteins from the same and other target families.

4) There are three main questions in this manuscript: the best computational modelling approaches, the optimal molecular and chemical descriptors and the most predictive bioactivity assays (?) and data sources. The study uses models that were submitted to the challenge, which hardly cover all possible modalities. The methods were tuned for specific targets, biased by personal preferences and there is no indication they can be more broadly applied, provided that data is available. There is also no comprehensive and systematic study on which descriptors provide the best results (I would have additional questions about the soundness of some models but it appears this is out of the authors' control. Therefore, I skip those comments). Moreover, there is promiscuity between the data sources, with several data entries being present in more than one database. It is not clear how the authors deconvolute this to conclude which datasets are the most promising for modelling. In line with my previous observation the third objective seems incorrectly formulated and would need rewording, at least. Based on these comments, I believe the study fails to answer the main research questions and deliver on its promises.

5) There is no clear reasoning as to why testing 430 small molecule–target relationships (round 1) in a space of 13930 combinations. Were all non-tested relationships previously reported in the literature? Minor comment: the units in the method description are wrong. It should be “ μM ” instead of “mM”.

6) The metrics for the grand majority of submitted models are far from impressive (Suppl Fig 4). Their distribution makes one wonder how imbalanced the dataset is. The authors should comment on that. Also, the authors' baseline model performs equally well compared to most models. All of the underperforming methods should be excluded from subsequent analyses as they only add noise to the message the manuscript tries to convey.

7) I do not see a reasonable motivation to discretise data (with an empirical hard cutoff) that is continuous by nature and build a classifier to extract an AUC. Also, depending on the data distribution (and the selected bins), ROC AUC might not be the best metric as it becomes optimistic in highly imbalanced sets. Other metrics should have been provided, such as balanced accuracy and precision-recall AUC.

8) There is no silver lining regarding the best models as initially questioned by the authors. Different methods perform equally well, but there is no interpretation of how they work and if the similar performance is connected to the identification of similar patterns. I assume this is not the case as an ensemble method seemingly performs better, which suggests that different models provide complementary vantage points on data. In the current shape, the manuscript does not add value to the machine learning community, but extracting this kind of information could be very interesting and useful for both cheminformaticians and med chemists.

9) It is confusing why the authors compare models built on pKd data with single dose experiments. The domain of applicability of the models is different and a skilled medicinal chemist will know the poor correlation between primary screen data and a full concentration-response curve. I do understand the authors try to provide an analysis as thorough as possible but this is not sound in my opinion. Therefore, I would remove this whole section, since the manuscript is targeted at medicinal chemists. Such a discussion will only raise unnecessary criticism.

10) One of my main concerns in this study is the quality/novelty of the chemical matter and the authors' data exposition in the last section. Please provide chemical structures to the highlighted molecules— TPKI-30 and GSK1379763. While I was able to google the structure of the GSK compound quite easily, I was only able to access and analyse the structure of TPKI-30 by coding a short snippet mining the authors' files. This will not be a viable option for medicinal chemists who want to take advantage of the findings. I can't also find an activity value for these two compounds, which should be straightforward taking into account that they are highlighted. Additionally, identifying TPKI-30 as a PYK2 inhibitor is not as challenging and surprising as the authors suggest. By extracting PYK2 data from ChEMBL, calculating MACCS keys for all compounds, performing a similarity search and scoring those searches with Tanimoto values, I was able to identify 4 molecules (ChEMBL1968380, ChEMBL513909, ChEMBL464552, ChEMBL3039525) with values above the similarity threshold (>0.80) and potent bioactivity/affinity values (as low as 76 nM). Also, one can find in ChEMBL more than 1880 molecules that have been studied against PYK2. By no means this target qualifies as under-studied – as mentioned by the authors – and the novelty of TPKI-30 is questionable. Similar conclusions can be taken for DDR1. This is a highly studied target but in this particular case a Tanimoto value of only 0.53 relative to the nearest neighbour in ChEMBL data was found. I did not perform an extensive literature search, but as it stands, the GSK compound could represent a novel chemotype for inhibition of DDR1.

11) Related to my previous comment, I am missing a comparison with other tools and methods that are more established at the target identification task. Could all or some of these findings be obtained with SEA or similarity searches? Possibly even molecular docking or pharmacophore screening? All these are more established tools and a superior performance of the method described herein could imply opportunities in early discovery chemistry.

12) I am missing a thorough validation of target binding and engagement. What is the concentration of ATP used in the assay? Multiple orthogonal assays should be used to confirm that the obtained bioactivity or affinity data is not artefactual.

13) The authors mention there are no systematic and unbiased evaluations applied to comprehensive datasets. This is unfortunately not true and overlooks important work performed by colleagues (e.g. *Chemical Science* 2018, 9, 5441). Likewise, I am not surprised that K_i , IC_{50} and EC_{50} data can be used to predict K_d values as there is also extensive validation of that procedure in the literature (e.g. *Angewandte Chemie International Edition* 2014, 53, 4244).

14) The conclusions are too overarching by suggesting these models can identify secondary pharmacology (not supported by data and beyond a subset of kinases), and prediction of selective compounds (not a single example in this manuscript). These conclusions should be greatly toned down.

Reviewer #3 (Remarks to the Author):

The manuscript by Wennerberg, Guinney, Aittokallio and co-workers reports on a competition for implementing machine learning models to predict bioactivity values of small molecules against several kinases. This is relevant research as it might enable the adoption and standardization of certain workflows in this space, while identifying potential pitfalls in model building practices and available chemical/biological data. In spite of this, I did find the message of the manuscript confusing, which made it difficult to fully gauge the significance of the results and methods. Moreover, the authors identify many research questions, but unfortunately, their data is not exceptional at addressing either of them. Overall, the findings in this study (including their validation and advances to state-of-the-art) are publishable, but fall short of what is expected from a Nature Communications manuscript.

Our response: We thank the Reviewer for appreciating the relevance of the Challenge, and are pleased to see the Reviewer found the results publishable. We have now improved the work according to the expert comments, and hope the revised manuscript will meet the expectations of the Reviewer and the general readership of Nature Communications. Please find our point-by-point responses below.

Below I provide a series of comments that support my assessment.

1) It is confusing how the predictive accuracy of a given model or ensemble of models can exceed that of a kinase activity assay. An activity assay does not predict activity but provides activity data. Whether the obtained value is accurate or not is a different question but, as it stands, the comparison is not correct.

Our response: We apologize that the original text was confusing in terms of kinase inhibition assays. We generated unpublished compound-target bioactivity data for unbiased scoring of the machine learning model predictions. More specifically, we sent the kinase inhibitors to DiscoverX (Eurofins Corporation) for the generation of new dose-response dissociation constant (K_d) values, as a measure of a compound-kinase binding affinity (please see Methods, page 25). These novel and comprehensive bioactivity data between 95 compounds and 295 kinases were then used as “ground truth” to evaluate the compound-target affinity prediction models submitted to the Challenge in terms of six evaluation metrics (root-mean-square error (RMSE), Pearson correlation, Spearman correlation, concordance index (CI), F1 score, and average AUC; see *Scoring of the model predictions* Methods section, page 27). In that sense, the predictive accuracy of a given model or ensemble of models cannot exceed that of the K_d measurements, rather the model predictions were compared against the ground truth K_d measurements using the various metrics.

The accuracy and robustness of the large-scale K_d bioactivity measurements generated for the Challenge were assessed by comparing them against existing K_d measurements from DiscoverX for the 412 compound-kinase pairs that overlapped with the Round 2 kinases that have been published in two comprehensive target profiling studies (from refs. 1 and 5). The correlation of replicate pK_d values was 0.842 (Supplementary Figure 25), hence demonstrating a good reproducibility of the multi-dose K_d measurements (page 25). These replicate measurements were also used when determining a practical upper limit for the predictive accuracy of the submitted prediction models and when scoring of the teams' predictions (see Figure 3b,d), where we also made use of several methods for statistical assessment to account for variability in the large-scale bioactivity measurements (please see Methods, pages 28-29).

The compounds included in the Challenge test data were a part of the kinase inhibitor collection at the SGC-UNC for which we already had the single-dose %inhibition screening done at DiscoverX across their large kinase panel. Such a two-step screening approach has been widely-used in previous studies (e.g., refs. 1-4), using the DiscoverX KINOMEScan standard protocol, where the multi-dose K_d values are generated only for compound-kinase pairs that had inhibition above 80% in the single-dose assay. After selecting the Challenge top-performing prediction models, we next

investigated how well these models compare against the single-dose activity assays when selecting most potent compound-target activities for the more detailed, multi-dose K_d profiling (see *Comparison against single-dose activity section* on page 15). For this classification task, we also defined the ground truth activity classes based on the measured K_d potencies, against which we compared both the single-dose activity data and the predictions from the ensemble of the top-performing models using discrimination analyses based on both receiver operating characteristic (ROC) curves and precision-recall (PR) curves (see Figure 7). We have also now added the balanced accuracy, as was requested by the Reviewer (comment 7 below).

References

1. Davis MI, et al. Comprehensive analysis of kinase inhibitor selectivity. *Nature Biotechnology* 2011;29(11):1046-1051. doi:10.1038/nbt.1990.
2. Elkins JM, et al. Comprehensive characterization of the Published Kinase Inhibitor Set. *Nature Biotechnology* 2016;34(1), 95-103. doi:10.1038/nbt.3374.
3. Drewry DH, et al. Progress towards a public chemogenomic set for protein kinases and a call for contributions. *Plos One* 2017;12(8), e0181585. doi:10.1371/journal.pone.0181585.
4. Karaman MW, et al. A quantitative analysis of kinase inhibitor selectivity. *Nature Biotechnology* 2008;26(1):127-132. doi:10.1038/nbt1358.
5. Fabian MA, et al. A small molecule-kinase interaction map for clinical kinase inhibitors. *Nature Biotechnology* 2005;23(3):329-336. doi:10.1038/nbt1068.

We observed that the combination of the top-performing machine learning models improved both the sensitivity and specificity of the target classifications, compared to using data from single-dose target activity assays, without requiring any additional experiments. This demonstrated that the machine learning models could be used in place of the single-dose measurement experiments when selecting compound-kinase pairs for multi-dose K_d profiling. However, we also demonstrated how combining the single-dose measurements with the prediction models led to even higher prediction accuracy than using either one alone; for instance, the combination of prediction models and single-dose measurements identified an increased number of potent compound-kinase activities compared to using the standard 80% inhibition cut-off (refs. 1-4; see Figure 8). We have now clarified the use of the two kinase inhibition assays in the revised manuscript (pages 15, 17, 22, and 25).

2) I understand that regression models were built to predict affinities, yet the authors mention that some of the activities found in this study were unexpected even for under-studied targets. From a medicinal chemistry and even machine learning point of view this does not make sense. If regression models were built, then a certain amount of data is present; thus the targets have been studied to a considerable extent. It would be different if unsupervised learning had been employed, but this is not the case.

Our response: It is indeed true that there were training bioactivity data available for all the kinases and compounds used in the Challenge for the teams to develop supervised learning models. However, the amount of bioactivity data available for different kinases varied considerably (see Figure R10 in the response to comment 5 of Reviewer #2), which was something that the supervised prediction models had to deal with in the model development and training phase. Importantly, no K_d measurements were available for the compound-kinase pairs that were used in the Challenge test phase, and for which we generated novel K_d data, as mentioned above.

We have now further analysed how much training data are needed for accurate affinity predictions (please see our responses to the Reviewer #2 comment 5 above). It is naturally subjective what is considered unexpected or under-studied, but these additional analyses show that the models can accurately predict binding affinities even for kinases such as PYK2, CSNK2A2 and DDR1 with a

relative small number of training bioactivities compared to other kinase targets in the Challenge. We have now described these new results in the revised manuscript (pages 17-18 and 20, 23).

3) The authors mention that chemical agents inhibiting single protein targets are rare and that a comprehensive understanding of pharmacological effects is important. The study does not seem to answer this need as all models are focused on a subset of kinases, and ignoring several potentially important interactions with proteins from the same and other target families.

Our response: It is correct that this Challenge focused on kinase inhibitors, since kinases are tractable in drug development and play a role in a wide range of diseases, including cancer, cardiovascular disorders and central nervous system diseases. Kinase activity prediction was also chosen for the Challenge because protein kinase domains share structural and sequence similarity, and most kinase inhibitors bind to conserved ATP-binding pockets, which leads to prevalent target promiscuity and polypharmacological effects. Therefore, we feel that kinase inhibitors provide a medically important, pharmacologically realistic, and computationally rather challenging target class for the multi-target predictive modelling (as mentioned on pages 3 and 5).

We also note that the K_d assays spanned a total of 95 compounds and 295 kinases (ca. 57% of the human protein kinome), which covered all the major kinase families and groups (Figure 2). The affinity data also represented many promiscuous compounds targeting multiple kinases at low concentrations, along with compounds with more narrow target profiles, as well as compounds with no potent targets among the tested kinases (please see Supplementary Figure 2). We therefore believe these novel Challenge Round 1 and 2 bioactivity datasets provided a standardized and sufficiently large quantitative bioactivity resource to evaluate the practical accuracy of predicting on/off-target activities of kinase inhibitors with relatively wide polypharmacological effects. This is now described in the revised version (pages 4-5).

It was one of the objectives of the Challenge to evaluate how well the predictive models can deal with such compound promiscuity that requires effective target deconvolution approaches. All the top-performing teams used machine learning models that can leverage the information extracted from similar kinases and/or compounds to predict the activity of so far unexplored interactions (see Table 1 and new Supplementary Table S1). Furthermore, most of the teams have used amino acid sequences or K-mer counting as target-based features in their class-agnostic prediction models. We therefore believe the Challenge models and the current benchmarking results will provide useful information for constructing predictive models also beyond kinases inhibitors.

We actually noticed that many teams used training bioactivity data for the number of distinct proteins exceeding the size of the human protein kinome, and therefore we have now also studied whether inclusion of bioactivities from other than protein kinase families into the model training leads to an improved compound-kinase K_d prediction accuracy. In particular, we compared the results between the original top-performing Q.E.D model that was based on protein kinases only and a version of the Q.E.D model that additionally incorporated training bioactivity data extracted from Drug Target Commons for G-protein-coupled receptors (GPCRs), ion channels, nuclear receptors, transcription factors and transporter proteins (ca. 100 000 compounds, out of which ca. 10 000 were kinase inhibitors). This increased the number of protein targets in the training dataset from roughly 530 to 900, including both kinases and other proteins, but the number of compounds in the prediction model was kept similar as in the original submitted model by selecting ca. 13 600 of 100 000 compounds most similar (based on Tanimoto similarity of Morgan fingerprints) to the ones in the Challenge test data (4 197 kinase inhibitors and 9 436 other compounds). The results showed that inclusion of other than kinase proteins and kinase inhibitors in the training data actually led to a significantly decreased prediction performance compared to the original model with kinases only (Pearson correlation of 0.482 vs. 0.557, $p=0.001$, Pearson and Filon test; Figure R15 below). This new result is mentioned in the text (page 22; and new Supplementary Figure 26).

Figure R15. Scatter plots between 394 measured and predicted Round 2 pK_d values based on the original Q.E.D model that included only kinases in the training data (red) and the new Q.E.D model that includes bioactivities (K_d , K_i and EC_{50}) for kinases and other proteins (GPCRs, ion channels, nuclear receptors, transcription factors and transporter proteins).

In the revised manuscript, we have also investigated how well the Challenge machine learning models predicted various kinase classes (among those present in the Challenge) to study their applicability ranges in more detail (please see page 14-15 and new *Class enrichment analyses* section in Methods on page 33). For example, the Challenge models provided accurate predictions across various kinase families, in general, but when considering all the 90 models with average absolute error below 2 over the 394 compound-kinase pairs in Round 2, the pairs involving MAPK and PDGFR kinases showed poorer accuracies compared to other kinase families ($p=0.001$, Kruskal-Wallis test). However, activities in these kinase families were better predicted using the top ensemble and the Q.E.D models (new Supplementary Figure 13, shown below in Figure R16). For the MAP kinases, the higher prediction errors (Benjamini-Hochberg adjusted $p=0.016$, Kolmogorov-Smirnov test) could be attributed to the fact that most of the inhibitors targeting MAP kinases are non-competitive allosteric inhibitors (1).

Similarly, pairs in the CMGC kinase group showed an increased error for most of the submissions in Round 2 (Benjamini-Hochberg adjusted $p=0.030$, Kolmogorov-Smirnov test), but again both the ensemble and Q.E.D models better predicted kinase activities in CMGC group (new Supplementary Figure 14, shown below in Figure R17). These results are added to the revised manuscript (pages 14-15).

References

1. Zhao Y, Adjei AA. The clinical development of MEK inhibitors. *Nature Reviews Clinical Oncology* 2014;11(7):385-400. doi:10.1038/nrclinonc.2014.83.

Figure R16. Kinase family enrichment analysis results. The overall p -values in the title of each plot were calculated based on Kruskal-Wallis test, and the group-specific Benjamini-Hochberg-adjusted p -values (only significant p -values, in red, are displayed) were calculated based on a weighted Kolmogorov-Smirnov-like statistic, similar to gene set enrichment analysis (see Methods for details). The values at the top of each plot indicate the number of compound-kinase pairs in each kinase family in the Round 2 dataset. The first panel considers 90 out of 99 Round 2 submissions with average absolute error below 2 over the 394 pairs in Round 2, based on which median absolute error was calculated and visualized.

Figure R17. Kinase group enrichment analysis results. The p-values on top of each panel were calculated based on Kruskal-Wallis test, and the group-specific Benjamini-Hochberg-adjusted p-values (only significant p-values, in red, are displayed) were calculated based on a weighted Kolmogorov-Smirnov-like statistic, similar to gene set enrichment analysis (see Methods for details). The n values indicate the number of compound-kinase pairs in each kinase group in the Round 2 dataset.

In the revised discussion, we also acknowledge the fact that the current dataset does not cover all the potentially important interactions with proteins from the same and other target families, and that future studies are warranted to extend the modeling to other target classes (pages 23-24).

4) There are three main questions in this manuscript: the best computational modelling approaches, the optimal molecular and chemical descriptors and the most predictive bioactivity assays (?) and data sources. The study uses models that were submitted to the challenge, which hardly cover all possible modalities. The methods were tuned for specific targets, biased by personal preferences and there is no indication they can be more broadly applied, provided that data is available. There is also no comprehensive and systematic study on which descriptors provide the best results (I would have additional questions about the soundness of some models but it appears this is out of the authors' control. Therefore, I skip those comments). Moreover, there is promiscuity between the data sources, with several data entries being present in more than one

database. It is not clear how the authors deconvolute this to conclude which datasets are the most promising for modelling. In line with my previous observation the third objective seems incorrectly formulated and would need rewording, at least. Based on these comments, I believe the study fails to answer the main research questions and deliver on its promises.

Our response: Like elaborated above (please see our response to comment 2 of Reviewer #2), the prediction of compound-target activities is indeed an active area of research, and there is an increasing number of new studies published each year that introduce novel computational algorithms to predict compound-target activities (see Figure R7, above). Therefore, it is practically impossible to have all the different prediction algorithm modalities covered in a comparative study. However, we feel that the total number of 268 prediction sets which were scored during the Challenge, and the wide range of predictive approaches, including regularized regression, deep and kernel learning algorithms, and gradient boosting decision trees, provided a representative set of prediction algorithms used in the field (see Table 1 and new Supplementary Table S1). We also expect that the implemented benchmarking infrastructure will support the future development and comparison of existing and emerging target activity prediction methods on the common dataset (Figure 1).

To support this statement, we have now also performed a systematic PubMed literature scan that revealed that most of the machine learning approaches used in the literature for drug-target binding affinity prediction are very well represented in the Challenge (new Figure R8 above, included as new Supplementary Figure 21). To date, there has been no systematic evaluation of these algorithms in common, blinded and large-enough data sets. This was the motivation for organizing the IDG-DREAM Challenge: to provide the community with systematic benchmarking results among various model classes, chemical features and data types. It is true that the methods were trained for specific kinase targets, but like noted above, we feel these comprehensive and promiscuous compound-kinase interactions provide an ideal testbench for the model testing.

Regarding the three study questions, please see our detailed responses to the comment 1 of Reviewer #2 above, including a series of new experiments to study which descriptors provide the best results. Furthermore, we have now also better clarified how the models could be used in future studies, both for additional sets of kinases, as well as for other target classes (pages 23-24). We note that the models were not specifically developed for the subset of kinases only, rather all the kinases available in the training data were used holistically in the model development, prediction and Challenge evaluation process, which is likely to make the application of the rather generic models more straightforward for other target and compound classes in the future. We also note that the open-source algorithms together with the novel bioactivities between 95 compounds and 295 kinases provide a resource for benchmarking new kinase inhibitor prediction algorithms.

It is true that there is promiscuity between the bioactivity data sources. However, our main focus in the comparison of data sources was the difference between various assays (e.g., multi-dose K_d , K_i , IC_{50} , EC_{50} and single-dose inhibition and potency assays), rather than comparison between various drug/target databases in terms of their coverage/accuracy for predictive modelling. The short comparison between ChEMBL, DTC and other data resources was originally included in the discussion, but we have now deleted this comparison in response to this constructive comment. We have now also deleted the “and data sources” part from the revised introduction (page 3), to make it clear that the Challenge aimed to find bioactivity assays (not data sources) that could help prediction of K_d values. With the new detailed results, we hope we have addressed the three study questions in a more systematic manner. Even if drastic differences are not always present, this result is also an important information for the future modelling and predictive studies for the community.

5) There is no clear reasoning as to why testing 430 small molecule–target relationships (round 1) in a space of 13930 combinations. Were all non-tested relationships previously reported in the literature? Minor comment: the units in the method description are wrong. It should be “ μ M” instead of “mM”.

Our response: Round 1 test dataset was generated based on a widely used two-step large-scale screening approach, where a single-dose kinome activity scan was first performed, and then the results of the single-dose activity scan were used to prioritise 430 most potent compound-kinase pairs for a dose-response determination of K_d values. Before the Challenge, we made sure that K_d values for these 430 compound-kinase pairs between 70 inhibitors and 199 kinases were not available in the public domain. We have now clarified this in the revised manuscript (page 4), as well as corrected the unit in the method description (page 25). Thank you for pointing this out.

6) The metrics for the grand majority of submitted models are far from impressive (Suppl Fig 4). Their distribution makes one wonder how imbalanced the dataset is. The authors should comment on that. Also, the authors' baseline model performs equally well compared to most models. All of the underperforming methods should be excluded from subsequent analyses as they only add noise to the message the manuscript tries to convey.

Our response: Since K_d data are continuous, we think the data is not imbalanced, but rather the regression problem is challenging due to the wide polypharmacological effects and non-uniform K_d distribution both in the test and training data (see Figure 2c, Supplementary Figure 2 and new Figure R10 included as new Supplementary Figure 20). All the underperforming models have indeed been ignored in the follow up analyses. Only the top-performing models (including Q.E.D, DMIS_DK, AI Winter is Coming) were subjected to further analyses and the ensemble modelling. We note that there were several models that outperformed our baseline model in both of the Challenge Rounds in terms of all the six evaluation metrics (see Supplementary Figure 4). We also note that the baseline model is not just any simple baseline reference model, but a method we developed and experimentally validated over >2 years of active research. We therefore find it quite impressive that so many models were able to outperform the baseline, given that the Challenge ran over a period of <6 months and many of the teams did not have any prior expertise in cheminformatics. For a more detailed comparison of the baseline model and the top-performing Q.E.D model, please see our response to comment 4 of Reviewer #2. We have added a note on how the unbalanced activity class distribution was treated in the Methods section (page 28).

7) I do not see a reasonable motivation to discretise data (with an empirical hard cutoff) that is continuous by nature and build a classifier to extract an AUC. Also, depending on the data distribution (and the selected bins), ROC AUC might not be the best metric as it becomes optimistic in highly imbalanced sets. Other metrics should have been provided, such as balanced accuracy and precision-recall AUC.

Our response: All the teams developed regression models for predicting continuous pK_d values, as was stated in the Challenge rules. Like noted above, the predictions were scored based on six different metrics that compare the measured and predicted pK_d values, out of which only F1 score and averaged AUC use discretized representation of the K_d data (and average AUC actually uses ten interaction threshold values from the pK_d interval [6, 8] to binarize pK_d 's into ten class labels). We have provided the Challenge results in terms of all the metrics in Supplementary Figure 4 both in Round 1 and Round 2 datasets. We have now also highlighted the top-performing teams in Round 2 in the modified Supplementary Figure 4. To select the top-performing models, we used two metrics, RMSE and Spearman correlation, as these metrics address two practical questions: continuous prediction of quantitative affinity values (RMSE), and prediction of the activity ranking across all the compound-target pairs (Spearman correlation, see page 7). Furthermore, these metrics provided a rather complementary view of the performance of the models; for instance, they showed less correlated performance over the less-accurate models in Round 2 (Figure 3f). In general, as expected, the six evaluation metrics are more or less correlated (Supplementary Figure 5), and therefore we felt describing all the results with each metric would make the text and figures unnecessary complicated and redundant.

Most of our follow-up analyses were based on the two selected metrics, RMSE and Spearman correlation, and therefore the continuous activity values were not discretized, as suggested by the Reviewer. Only when in post-Challenge phase we went to a comparison against the single-dose assay measurements, we chose to use the ground truth activity classes based on the measured K_d values in the evaluation, since the single-dose inhibition measurements are in the range of 0-1, whereas the pK_d values range between 5-10, so we felt classification problem provides fairer comparison between predicted pK_d and measured %inhibition values. The classification analyses also provided a more practical prediction outcome, compared to the absolute error or rank correlation analyses in the Challenge phase that already demonstrated predictive power of the top-performing models. Here, the practical question was a classification task, that is, whether the compound-target pair should be selected for further K_d profiling or not. The use of classification analyses also offered the possibility to calculate additional evaluation metrics, such as sensitivity, specificity and positive predictive value (PPV), that may be easier to interpret than the overall RMSE, correlation or AUC values alone. The precision-recall (PR) curves and the corresponding PR-AUC values were already shown in original Figure 5B (new Figure 7B) and Supplementary Figure 9b (new Suppl. Fig. 15b) for comparison to the ROC curves and their respective AUC values in the original version.

As requested by the Reviewer, we have now also listed the balanced accuracy and PR-AUC values for all the classification analyses described in the manuscript (please see the values in parentheses in the modified Figures 7, and Supplementary Figures 15 and 17). These metrics are summarized in the Table R2 (see below), in comparison to the ROC-AUC values. In general, all these evaluation metrics support the benefits of the ensemble model, when compared to the Q.E.D model alone, and especially the benefits compared to the single-dose %inhibition measurements. For instance, in the classification analyses in Figure 7, the combined model (ensemble of top-4 prediction models and %inhibition measurements) obtained the highest accuracy in terms of all the metrics. Two different activity cut-offs (pK_d of 6 and 7) were tested to see how the active/inactive class sizes affect the results (see page 33). All the evaluation metrics are described in the Materials and Methods section (page 27-28).

Table R2. Balanced accuracy for the activity classification, compared to AUC values. The highest accuracies are boldfaced.

Figure	Number of pairs	Model	Activity cut-off*	Balanced accuracy	ROC-AUC	PR-AUC
Figure 7	Total: 394 Actives ($pK_d > 6$): 252 Inactives ($pK_d \leq 6$): 142	Inhibition%	>80%	0.55	0.64	0.70
		Ensemble	$pK_d > 6$	0.67	0.76	0.86
		Q.E.D.	$pK_d > 6$	0.64	0.73	0.83
Figure 8	Total: 475 Actives ($pK_d > 6$): 275 Inactives ($pK_d \leq 6$): 200	Inhibition%	>80%	0.64	0.70	0.70
		Ensemble	$pK_d > 6$	0.60	0.64	0.72
		Q.E.D.	$pK_d > 6$	0.58	0.60	0.65
		Combined	>80%	0.70	0.81	0.87
Suppl. Fig. 15	Total: 394 Actives ($pK_d > 7$): 116 Inactives ($pK_d \leq 7$): 278	Inhibition%	>80%	0.55	0.79	0.55
		Ensemble	$pK_d > 7$	0.67	0.76	0.59
		Q.E.D.	$pK_d > 7$	0.66	0.74	0.58

*Single-dose inhibition >80% is a standard cut-off in large-scale kinase inhibitor profiling studies (refs. 1-4). The cut-off for the predicted pK_d was set equal to the ground-truth activity classification based on the measured K_d . Two different activity cut-offs were tested to see how the class balance affects the results.

References

1. Davis MI, et al. Comprehensive analysis of kinase inhibitor selectivity. *Nature Biotechnology* 2011;29(11):1046-1051. doi:10.1038/nbt.1990.
2. Elkins JM, et al. Comprehensive characterization of the Published Kinase Inhibitor Set. *Nature Biotechnology* 2016;34(1), 95-103. doi:10.1038/nbt.3374.
3. Drewry DH, et al. Progress towards a public chemogenomic set for protein kinases and a call for contributions. *Plos One* 2017;12(8), e0181585. doi:10.1371/journal.pone.0181585.
4. Karaman MW, et al. A quantitative analysis of kinase inhibitor selectivity. *Nature Biotechnology* 2008;26(1):127-132. doi:10.1038/nbt1358.

8) There is no silver lining regarding the best models as initially questioned by the authors. Different methods perform equally well, but there is no interpretation of how they work and if the similar performance is connected to the identification of similar patterns. I assume this is not the case as an ensemble method seemingly performs better, which suggests that different models provide complementary vantage points on data. In the current shape, the manuscript does not add value to the machine learning community, but extracting this kind of information could be very interesting and useful for both cheminformaticians and med chemists.

Our response: It is not correct that different prediction methods performed equally well, rather we observed quite large differences in their predictive performance (see Figure 3 and Supplementary Figures 4-7). However, it is true that the 4 top-performing prediction models and their features provided complementary predictive contributions to the compound-kinase activities, and therefore the ensemble of the best-performing models was constructed and shown to result in the best overall performance. Even though this may seem as an unsatisfying outcome from the machine learner point of view, we actually think this is a nice demonstration of the “wisdom of crowds” concept, namely, that one can obtain more robust and accurate predictions by combining various model classes and features (as mentioned on pages 9-10 and 22).

In the revised manuscript, we have now listed the model features of all the teams, including the worse-performing teams, in the new Supplementary Table S1, similar to the Table 1 for best-performing models only, to provide further guidance on what are the ingredients of the top-predictive performance. Please also see above our detailed responses, supported by several new experiments, to the comments 1, 4 and 7 of Reviewer #2 to answer the comment of extracting information of the determinants of models' performance. We agree that these modelling insights are important for the cheminformaticians and medicinal chemists, who are interested in using predictive models to guide compound-target mapping efforts.

9) It is confusing why the authors compare models built on pK_d data with single dose experiments. The domain of applicability of the models is different and a skilled medicinal chemist will know the poor correlation between primary screen data and a full concentration-response curve. I do understand the authors try to provide an analysis as thorough as possible but this is not sound in my opinion. Therefore, I would remove this whole section, since the manuscript is targeted at medicinal chemists. Such a discussion will only raise unnecessary criticism.

Our response: This comparison was carried out because single-dose experiments are currently widely used as part of two-phase screening technique to map and validate large spaces of compound-target activities using single-dose and multi-dose compound-target assays, respectively. The correlation between the single-dose activity screen and the multi-dose K_d assay

was actually not too poor, but behaved as expected in the Challenge dataset (see Figure 8a). The comparison between the single-dose measurements and the use of prediction models highlights the utility of the machine learning models in selecting potential high affinity compound-target pairs for further evaluation and multi-dose profiling that may be missed by the single-dose screening experiments (see Figure 8b-e). We therefore would like to keep this comparison in the manuscript as we feel it is important not only for target profiling applications but also for medicinal chemists when selecting molecules for further development.

10) One of my main concerns in this study is the quality/novelty of the chemical matter and the authors' data exposition in the last section. Please provide chemical structures to the highlighted molecules– TPKI-30 and GSK1379763. While I was able to google the structure of the GSK compound quite easily, I was only able to access and analyse the structure of TPKI-30 by coding a short snippet mining the authors' files. This will not be a viable option for medicinal chemists who want to take advantage of the findings. I can't also find an activity value for these two compounds, which should be straightforward taking into account that they are highlighted. Additionally, identifying TPKI-30 as a PYK2 inhibitor is not as challenging and surprising as the authors suggest. By extracting PYK2 data from ChEMBL, calculating MACCS keys for all compounds, performing a similarity search and scoring those searches with Tanimoto values, I was able to identify 4 molecules (ChEMBL1968380, ChEMBL513909, ChEMBL464552, ChEMBL3039525) with values above the similarity threshold (>0.80) and potent bioactivity/affinity values (as low as 76 nM). Also, one can find in ChEMBL more than 1880 molecules that have been studied against PYK2. By no means this target qualifies as under-studied – as mentioned by the authors – and the novelty of TPKI-30 is questionable. Similar conclusions can be taken for DDR1. This is a highly studied target but in this particular case a Tanimoto value of only 0.53 relative to the nearest neighbour in ChEMBL data was found. I did not perform an extensive literature search, but as it stands, the GSK compound could represent a novel chemotype for inhibition of DDR1

Our response: TPKI-30 and GSK1379763 are novel compounds, hence their target activities were highlighted as novel findings in the original manuscript. As was noted above, this section of the manuscript investigates the effectiveness of prediction models for prioritising the compound-target pairs for comprehensive multi-dose screening in comparison to the widely-used single-dose %inhibition assay experiments (see Figures 8). As outlined by the Reviewer, verification of TPKI-30 as a potent PYK2 inhibitor is indeed not a challenging task, once the interaction has been first identified by other means, but what we argue in our work is that systematic identification of a compound with a low single-dose inhibition value (below 50% for TPKI-30-PYK2 interaction) and a high multi-dose pK_d value (here, 7.3) is a very cumbersome task that cannot be carried out systematically in practice without the use of prediction models. This section of the manuscript highlights how the top-prediction models, when used with or without the single-dose assays, can aid the selection of effective small-molecule binders of particular targets for further development.

Moreover, we have conducted the Reviewer's analysis for the two other interactions highlighted in the revised manuscript (GSK1379763-DDR1 and PFE-PKIS14-CSNK2A2). In both cases, no compounds found in ChEMBL exceeded the Tanimoto threshold of 0.8, or even 0.7, in the analysis related to PFE-PKIS14-CSNK2A2. In case of GSK1379763-DDR1, one compound (ChEMBL124660) exceeded the threshold of 0.75, but with a relatively high K_d of 1400 nM.

It is true that when one downloads all the data from ChEMBL or DTC and calculates the number of compounds that have been studied against PYK2, the number of unique molecules equals to 1880 as the Reviewer pointed out. However, 1543 of the 1880 data points are in a form of activity types other than those used in the present work (e.g., IC_{50} , EC_{50} , K_i , K_d). Further filtering out the inactive interactions reduces the number of potent interactions to 265. Our new Figure R10 (included in response to the comment 5 of Reviewer #2 and as a new Supplementary Figure 20) illustrates the activity-filtered compounds and interactions only, which we believe provides a more informative view of the training data available for each Challenge kinase target. Similarly, a total of 103 active interactions were identified in the bar corresponding to DDR1 in Figure R10. Please also see our

responses to the comment 5 of Reviewer #2 above. We agree that also GSK1379763 represents a novel chemotype for inhibition of DDR1 as highlighted in the manuscript (page 17).

We have now added to Figure 8 one more novel interaction between PFE-PKIS14 and CSNK2A2 ($pK_d=6.2$) that was predicted by the top-performing Q.E.D model with high accuracy despite rather low amount of training data available (Figure R10 and Table R1 included as new Supplementary Figure 20), and would have been missed by the single-dose activity assay given the standard 80% inhibition threshold (pages 17-18). We note that CSNK2A2 is one of the target “dark” kinases nominated by the IDG consortium (1), suggesting that the ML prediction models can identify potent inhibitors even for the currently under-studied kinases.

As requested, we have also provided the chemical structures of the highlighted compounds (TPKI-30, GSK1379763 and PFE-PKIS14) by using the SMILES and InChiKey structural format of the compounds (please see the new uncaptioned PDF figure for chemical structures). We have further provided the SMILES, InChiKeys, single-dose %inhibition values and multi-dose K_d values for all the compound-target pairs used in the Challenge in CSV files for Reviewers only. These data will become available for the whole community upon the manuscript publication (please see the Data availability statement, pages 34-35).

References

1. Berginski ME, et al. The Dark Kinase Knowledgebase: an online compendium of knowledge and experimental results of understudied kinases. *Nucleic Acids Research* 2020. doi:10.1093/nar/gkaa853.

11) Related to my previous comment, I am missing a comparison with other tools and methods that are more established at the target identification task. Could all or some of these findings be obtained with SEA or similarity searches? Possibly even molecular docking or pharmacophore screening? All these are more established tools and a superior performance of the method described herein could imply opportunities in early discovery chemistry.

Our response: The original manuscript focused on benchmarking of the models submitted to the Challenge (268 submissions in total), which cover a range of different learning algorithms for binding affinity predictions (see Table 1, new Supplementary Table S1 and new Supplementary Figure 21). Even though systematic comparison of computational models developed over the years for target identification is outside of the scope of the present work, we agree that comparison against an established method, such as SEA, is warranted to demonstrate the added value of the machine learning algorithms.

We applied the online SEA web-application (<http://sea.bkslab.org/search>) to make predictions for the three compounds highlighted in the revised manuscript, TPKI-30, GSK1379763 and PFE-PKIS14, for which Q.E.D model predicted strong activity against DDR1, PTK2B (PYK) and CSNK2A2 ($pK_d>6$), respectively (Figure 8), and which were subsequently experimentally validated. In the SEA method, we used ECFP4 fingerprints that were also used by the top-performing prediction models in the Challenge (see Table 1; page 34).

Strikingly, SEA did not predict target activity against any of the 3 selected kinases (see new Supplementary Table S2; the strongest SEA hits are also listed in the Table R3 below), even though these interactions were experimentally confirmed with high activity ($pK_d>6$). The highest scoring hit from SEA for TPKI-30 was PTK2 (focal adhesion kinase 1), that belongs to the same subfamily of FAK kinases as PTK2B (protein-tyrosine kinase 2-beta), that was confirmed as potent target of TPKI-30, but their sequence identity is only ~43%.

We have now described these new comparative results in the revised discussion (page 19). We have now also shown in a case example that the standard docking model is not informative enough for the pK_d prediction for the interaction between TPKI-30 and PYK2 that was predicted by the ensemble model and experimentally validated (please see our response to Reviewer #1

comment 3). This result is now also described briefly in the results (page 19). We also expect that the implemented benchmarking infrastructure will support the future development and comparison of existing and emerging target activity prediction methods on the common dataset (Figure 1).

Table R3. Round 2 targets predicted by SEA with high statistical support ($p < 10^{-15}$, SEA-suggested cut-off) for the three compounds highlighted in the manuscript (PFE-PKIS14, TPKI-30 and GSK1379763). The targets with $p < 10^{-10}$ are listed in the new Supplementary Table S2.

Compound	SEA-predicted target		
	Gene symbol	UniProt ID	P-value
PFE-PKIS 14	CHEK2	O96017	1.11e-16
TPKI-30	PTK2	Q05397	4.626e-121
	CDK4	P11802	3.747e-38
	PLK1	P53350	4.647e-37
	NEK3	P51956	4.752e-33
	ALK	Q9UM73	4.345e-29
	BMX	P51813	1.337e-24
	CSF1R	P07333	5.055e-21
	TTK	P33981	9.975e-21
	NIM1K	Q8IY84	1.463e-20
	PLK2	Q9NYY3	2.467e-19
GSK1379763	PTK2	Q05397	4.446e-98
	INSR	P06213	1.216e-58
	TBK1	Q9UHD2	4.823e-58
	SYK	P43405	1.92e-53
	KDR	P35968	2.235e-38
	ERBB2	P04626	7.395e-37
	PDGFRA	P16234	6.12e-35
	PDGFRB	P09619	1.648e-33
	MAPK8	P45983	8.435e-29
	LCK	P06239	3.774e-22
	MAPK9	P45984	5.033e-19
	SRC	P12931	5.848e-18
	AURKA	O14965	1.585e-17
	PLK1	P53350	1.11e-16
	ABL1	P00519	4.441e-16

12) I am missing a thorough validation of target binding and engagement. What is the concentration of ATP used in the assay? Multiple orthogonal assays should be used to confirm that the obtained bioactivity or affinity data is not artefactual.

Our response: As noted above, the accuracy and robustness of the validation K_d bioactivity measurements were assessed by comparing them against a subset of existing K_d measurements for the same set of inhibitors-kinase pairs. The correlation of replicate pK_d values was 0.842 (Supplementary Figure 25), hence demonstrating a good reproducibility of the K_d measurements in such large-scale analysis. We therefore believe these bioactivity data are not artefactual.

The KINOMEScan assay protocol at DiscoverX is actually not an ATP competitive binding assay, rather broad spectrum ligands (e.g., staurosporine) are biotinylated and immobilized using streptavidin-coated magnetic beads on a solid support. The assay measures the amount of kinase bound to the immobilized ligand in the presence and absence of the test compound. The

competition as such is between the immobilized ligands and test compound for the same binding pocket of a given kinase. Therefore, the assay used in our study is a competitive binding assay, but it does not use ATP cofactors and substrates for competition. Instead it uses a broad spectrum of ligands for the K_d estimation. This is now detailed in the revised Methods section (page 25).

13) The authors mention there are no systematic and unbiased evaluations applied to comprehensive datasets. This is unfortunately not true and overlooks important work performed by colleagues (e.g. Chemical Science 2018, 9, 5441). Likewise, I am not surprised that K_i , IC_{50} and EC_{50} data can be used to predict K_d values as there is also extensive validation of that procedure in the literature (e.g. Angewandte Chemie International Edition 2014, 53, 4244).

Our response: We thank the Reviewer for pointing out these two interesting studies. The first study (Chemical Science 2018, 9, 5441) compares binary classification models (FNN, SVM, kNN, NB, RF and SEA) in the activity classification using the discretized ROC-AUC as the scoring metric. We note that our Challenge included most of these and many other newer prediction models (see new Supplementary Table S1 and new Supplementary Figure 21). Importantly, as pointed out by the Reviewer above, bioactivity data are continuous in nature, so regression models are expected to be more informative than classification models, as they can predict the actual compound-target activity. Furthermore, while the ChEMBL database that was used in this comparison work is indeed comprehensive, and the cross-validations included several target classes, the study lacks any blinded evaluation using unpublished data or experimental validation of the model predictions, that we carried out in the Challenge. As training datasets, our Challenge participants used three large-scale databases, including ChEMBL, DTC and BindingDB, to obtain comprehensive training data for model development, instead of relying on a single data resource only. Similar to our study, they also observed that larger training set sizes led to better predictions, on average, but the increase in AUC was not too impressive after a sufficient number of training data was available. We have now cited this study in the revised Discussion (page 21).

The second study uses the concept of ant colony optimization to combinatorial building block selection as computational molecular de novo design method for constructing bioactive compounds with desired on- and off-target binding. The authors trained individual Gaussian process regression models for 640 human targets, based on measured bioactivities from ChEMBL database. Molecules were represented by topological pharmacophore ("CATS2") and substructure (circular Morgan fingerprints) descriptors. This study indeed made use of K_d , K_i and IC_{50} affinity data, but the actual impact of integrating different types of bioactivity data on the prediction accuracy was not investigated, unlike in our revised manuscript. Furthermore, we feel that the de novo design of compounds is outside of the scope of the present study, where we focus on kinase inhibitors already available, and therefore we did not cite this paper in the revised version.

14) The conclusions are too overarching by suggesting these models can identify secondary pharmacology (not supported by data and beyond a subset of kinases), and prediction of selective compounds (not a single example in this manuscript). These conclusions should be greatly toned down.

Our response: We have not used the term "secondary pharmacology" in the original manuscript. In the revised version, we have emphasized that the Challenge focuses on compound-kinase activities (many places of the text), and as suggested, have now toned down the conclusions (page 20).

Reviewers' Comments:

Reviewer #2:

Remarks to the Author:

I have carefully reviewed the responses of the authors to my critiques and I am satisfied that they have been fully addressed. I commend the authors for their thorough consideration of the points raised and updated supporting data presented. The revised manuscripts provide interesting new insights into how model features and training data composition impact prediction accuracy. As a result, I find the current manuscript much richer than the prior submission. I have a few remaining minor points that, from my perspective, should not hold up publication of this interesting article.

Minor points:

In the revised abstract I'm concerned that the statement "a large proportion of the human kinome remains undruggable" might be misinterpreted to mean that there is something inherently "undruggable" about a fraction of the kinome. Perhaps replacing "undruggable" with "as yet undrugged" would better reflect the author's intent.

In Fig 6, full amino acid sequences used as protein kernels performed significantly better than kinase domain sequences. I found this surprising assuming that "kinase domain sequences" reflects the sequence of the canonical ePK domain. Can the authors speculate on how non-conserved regions outside of the kinase domain contribute to the accuracy of the modeling?

"The triangle shape indicates..." missing "triangle" in the legend to Fig 3e,f

-Jeffrey Peterson

Reviewer #3:

Remarks to the Author:

I had originally provided feedback on the manuscript by Wennerberg, Guinney, Aittokallio and colleagues, and assessed its merits from a cheminformatics and medicinal chemistry perspective. While recognizing the value of the study, my initial assessment was against publication in Nature

Communications. This was based on concerns regarding the depth to which the authors answered their research questions, i.e. the manuscript covered a lot of ground but did not really advance kinase inhibitor discovery to a desirable extent.

Undeniably, the authors have put in an immense effort to address all concerns raised by the 3 reviewers, and in my opinion the provided answers are satisfactory for the most part. In addition, the ms/SI files were revised accordingly and now feature new figures and discussion.

I still have a few concerns. For example, the authors make the observation:

“Interestingly, we observed that while Kd alone or in combination with other bioactivity data types, especially with Ki, resulted in rather accurate Kd predictions, the other data types, alone or in combination, led to significantly worse prediction performances. Especially the most abundant EC50 and IC50 multi-dose bioactivities alone led to a much poorer Kd prediction accuracy which, in case of IC50, cannot be explained by the smaller number of training data points.”

This should not be particularly interesting and/or unexpected because EC/IC50 is not an affinity metric (such as Kd). EC/IC50 values are dependent on protein concentration in the assay and, therefore, can be tuned. If the authors make such a consideration in the manuscript they should include an interpretation beyond a data science perspective.

Although the manuscript is in much better shape now, my opinion regarding the novelty of insights still stands (or novelty from a ML point of view). However, given the interesting Competition, the remarkable volume of data and enhanced discussion I would not oppose publication in Nature Communications.

Reviewer #2 (Remarks to the Author):

I have carefully reviewed the responses of the authors to my critiques and I am satisfied that they have been fully addressed. I commend the authors for their thorough consideration of the points raised and updated supporting data presented. The revised manuscripts provide interesting new insights into how model features and training data composition impact prediction accuracy. As a result, I find the current manuscript much richer than the prior submission. I have a few remaining minor points that, from my perspective, should not hold up publication of this interesting article.

Minor points:

In the revised abstract I'm concerned that the statement "a large proportion of the human kinome remains undruggable" might be misinterpreted to mean that there is something inherently "undruggable" about a fraction of the kinome. Perhaps replacing "undruggable" with "as yet undrugged" would better reflect the author's intent.

Our response: We agree and have replaced "undruggable" with "as yet undrugged" in the abstract (page 2).

In Fig 6, full amino acid sequences used as protein kernels performed significantly better than kinase domain sequences. I found this surprising assuming that "kinase domain sequences" reflects the sequence of the canonical ePK domain. Can the authors speculate on how non-conserved regions outside of the kinase domain contribute to the accuracy of the modeling?

Our response: We agree that this is a rather surprising observation. The reason is most likely due to the number of missing kinase domain sequences. Specifically, the Q.E.D model resulted in several pK_d predictions of zero (7%) for compound-kinase pairs for which amino acid sequences of kinase domains were not available. We have now confirmed that Pearson correlation of the model with kinase domain sequences increased markedly when the predictions of zero were omitted in the calculation (0.37 vs. 0.46), but it still remains lower than the correlation of the Q.E.D model with the full amino acid sequences. This is most likely due to the model having more training data available, since no compound-kinase pairs were excluded based on sequence availability. This is now briefly noted on page 14.

"The triangle shape indicates..." missing "triangle" in the legend to Fig 3e,f

Our response: Thank you for pointing this out, we have added the missing "triangle" word in the caption of Figure 3 (page 34). We note that the triangle shapes indicating DL methods are only present in panel f of Figure 3 (Round 2 submissions), because the information of method classes was not yet available for Round 1 submissions (panel e).

-Jeffrey Peterson

Reviewer #3 (Remarks to the Author):

I had originally provided feedback on the manuscript by Wennerberg, Guinney, Aittokallio and colleagues, and assessed its merits from a cheminformatics and medicinal chemistry perspective. While recognizing the value of the study, my initial assessment was against publication in Nature Communications. This was based on concerns regarding the depth to which the authors answered their research questions, i.e. the manuscript covered a lot of ground but did not really advance kinase inhibitor discovery to a desirable extent.

Undeniably, the authors have put in an immense effort to address all concerns raised by the 3 reviewers, and in my opinion the provided answers are satisfactory for the most part. In addition, the ms/SI files were revised accordingly and now feature new figures and discussion.

Our response: We are grateful for the positive comments and continued efforts to improve our work.

I still have a few concerns. For example, the authors make the observation:

“Interestingly, we observed that while Kd alone or in combination with other bioactivity data types, especially with Ki, resulted in rather accurate Kd predictions, the other data types, alone or in combination, led to significantly worse prediction performances. Especially the most abundant EC50 and IC50 multi-dose bioactivities alone led to a much poorer Kd prediction accuracy which, in case of IC50, cannot be explained by the smaller number of training data points.”

This should not be particularly interesting and/or unexpected because EC/IC50 is not an affinity metric (such as Kd). EC/IC50 values are dependent on protein concentration in the assay and, therefore, can be tuned. If the authors make such a consideration in the manuscript they should include an interpretation beyond a data science perspective.

Our response: We agree and have included such interpretation in the revised text (page 8).

Although the manuscript is in much better shape now, my opinion regarding the novelty of insights still stands (or novelty from a ML point of view). However, given the interesting Competition, the remarkable volume of data and enhanced discussion I would not oppose publication in Nature Communications.